# Molecular and functional heterogeneity in dorsal and ventral oligodendrocyte progenitor cells of the mouse forebrain in response to DNA damage

Enrica Boda [1,2✉], Martina Lorenzati [1,2], Roberta Parolisi [1,2], Brian Harding[3], Gianmarco Pallavicini [1,2], Luca Bonfanti [2,4], Amanda Moccia [5], Stephanie Bielas [5], Ferdinando Di Cunto [1,2] & Annalisa Buffo [1,2]

In the developing mouse forebrain, temporally distinct waves of oligodendrocyte progenitor cells (OPCs) arise from different germinal zones and eventually populate either dorsal or ventral regions, where they present as transcriptionally and functionally equivalent cells. Despite that, developmental heterogeneity influences adult OPC responses upon demyelination. Here we show that accumulation of DNA damage due to ablation of citron-kinase or cisplatin treatment cell-autonomously disrupts OPC fate, resulting in cell death and senescence in the dorsal and ventral subsets, respectively. Such alternative fates are associated with distinct developmental origins of OPCs, and with a different activation of NRF2-mediated anti-oxidant responses. These data indicate that, upon injury, dorsal and ventral OPC subsets show functional and molecular diversity that can make them differentially vulnerable to pathological conditions associated with DNA damage.

[1] Department of Neuroscience Rita Levi-Montalcini, University of Turin, Turin, Italy. [2] Neuroscience Institute Cavalieri Ottolenghi (NICO), University of Turin, Regione Gonzole 10, IT-10043 Orbassano (Turin), Italy. [3] Pathology and Laboratory Medicine, Perelman School of Medicine, University of Pennsylvania and Children's Hospital of Philadelphia, Philadelphia, PA, USA. [4] Department of Veterinary Sciences, University of Turin, Turin, Italy. [5] Department of Human Genetics, University of Michigan, Ann Arbor, MI, USA. ✉email: enrica.boda@unito.it

Vertebrate neurons exhibit an enormous diversity in morphology, neurochemical profile, function, and susceptibility to injury. The extent of this diversity is instead less well-established for neuroglia cells. Increasing evidence supports a high degree of developmental, molecular, and morpho-functional heterogeneity for astrocytes[1–4]. Recent studies have also identified distinct subsets of mature oligodendrocytes (OLs) in the human and mouse Central Nervous System (CNS)[5–8]. The diversity of oligodendrocyte progenitor cells (OPCs) is instead under debate[9], although many lines of evidence indicate that OPCs are not a homogeneous population, but rather comprise subsets endowed with distinct properties[10–12]. One major aspect of OPC diversity is their developmental origin. In both the brain and spinal cord, OPCs arise as temporally distinct waves from different ventral and dorsal domains of the ventricular germinal zones (VZ)[13]. In the mouse forebrain, the first OPCs are generated around E12.5 from Nkx2.1/Dbx1-expressing ($^+$) progenitors located in the medial ganglionic eminence (MGE) and anterior entopeduncular (AEP)/preoptic area (POA) of the ventral telencephalon. The second wave of OPCs arises around E15.5 from Gsh2$^+$ progenitors in the lateral and caudal ganglionic eminences. These embryonically generated OPCs initially colonize the entire brain. Finally, around birth, OPCs start to be generated by Emx1$^+$ progenitors in the dorsal VZ. Dorsally derived OPCs progressively replace the Nkx2.1-derived OPCs in the cortex and corpus callosum (CC), becoming the dominant population at these sites[14–21]. Such ventro-dorsal sequence of OPC generation is conserved also in the human brain[22–25].

Yet, at postnatal stages, ventrally and dorsally derived mouse OPCs show similar electrophysiological properties[11,26], as well as convergent transcriptional profiles[11]. Moreover, when one of these two subsets is experimentally ablated during development, the persisting population expands and compensates for the OPC loss[14,27,28]. Thus, OPCs with distinct origins appear functionally equivalent. However, data suggest that developmental heterogeneity may influence OPC behavior during aging and their susceptibility/response to injury. In adult mice, ventrally derived OPCs appear more resilient to lysolecithin-induced injury compared to dorsally derived OPCs[29]. However, these latter cells show a more efficient response to demyelination, although they also undergo a more marked age-associated reparative decline, compared to their ventral counterparts[29]. After perinatal hypoxia-ischemia, the regenerative response of striatal OPCs is more robust than that of cortical cells[30], whereas OPCs in the cortex are primarily affected and undergo depletion upon vanadium-induced developmental intoxication[31,32]. Thus, OPC developmental diversity may contribute to different regional manifestations of de-/dys-myelinating diseases.

Such functional diversity may arise, at least in part, from cell-intrinsic differences[29], whose molecular bases are still unknown. We addressed this possibility by studying a mouse model of human microlissencephaly, in which the germinal ablation of Citron-kinase (Cit-k, a cytoskeleton regulator involved in cell division and DNA repair[33,34]) resulted in profound myelination defects and triggered distinct responses in telencephalic dorsal (dOPCs; i.e., populating the dorsal cortex and corpus callosum, and derived from Emx1$^+$ progenitors) and ventral OPCs (vOPCs; i.e., ventrally derived and located in the striatum and hypothalamus). Both populations showed high levels of DNA damage. However, Cit-k KO dOPCs underwent depletion, whereas vOPCs persisted and displayed a senescent phenotype. Such differential sensitivity was observed also when Cit-k was deleted only in oligodendroglia or in either Emx1$^+$ or Nkx2.1$^+$ precursors and derivatives, indicating that cell-intrinsic factors associated with OPC developmental origin underlie Cit-k KO dOPC vs. vOPC alternative fates. Mechanistically, the two Cit-k KO OPC subsets showed distinct accumulation of reactive oxygen species (ROS) and a different ability to set up Nuclear factor erythroid 2-related factor 2 (NRF2)-mediated anti-oxidant defenses. Distinct vulnerability accompanied by diverse accumulation of ROS and expression of NRF2 were also observed in wild-type (WT) dOPCs vs. vOPCs upon cisplatin-induced DNA damage. These data provide novel evidence of functional and molecular heterogeneity in OPC subsets and indicate that distinct forebrain OPC populations may be differentially vulnerable to CNS disorders associated with DNA damage.

## Results

**CIT-K loss is associated with DNA damage and profound myelination defects in both mouse and human CNS.** Consistent with CIT-K participation in DNA damage repair[34], widespread phospho-histone H2AX (γH2AX) labeling, a marker of DNA breaks[35], was detected throughout Cit-k KO mouse forebrain at postnatal day (P) 14 (Fig. 1a, b). In humans, loss-of-function mutations in CIT-K are among the genetic bases of primary microlissencephaly leading to perinatal lethality (MCPH17)[36,37]. DNA damage was also detected in the cerebral cortex of an individual with MCPH17 due to biallelic truncating variants of CIT-K (CIT-K fs/fs; Fig. 1c, d).

Myelin and oligodendrocyte alterations were detected in both Cit-k KO brain and CIT-K fs/fs post-mortem cortical tissue. The forebrain and cerebellum of Cit-k KO mice lacked both premyelinating and fully differentiated oligodendrocytes, as shown by almost absent expression of the premyelinating stage marker GPR17[38] and of myelin proteins (PLP1, MBP) (Fig. 1e, f, k–p and Supplementary Fig. 1a). Gallyas silver staining (Fig. 1q–v) and electron microscopy investigations (Fig. 1y, z) confirmed the absence of myelin throughout the whole Cit-k KO forebrain and spinal cord, despite the presence of numerous axons (Fig. 1w, x). Likewise, severe hypomyelination of subcortical and cerebellar white matter was observed in CIT-K fs/fs tissue (Fig. 1g–j).

**Cit-k loss differentially affects dorsal and ventral OPCs in the postnatal mouse forebrain.** OPCs were evaluated to investigate the origin of the myelination deficit in Cit-k KO mouse brain. At P3, NG2$^+$ OPCs were reduced by half in the dorsal cortex and striatum of Cit-k KO, compared to WT controls (Fig. 2a, b, d). Consistently, post-mortem quantifications in the deepest layers of the cerebral cortex of CIT-K fs/fs MCPH17 subject showed a 50% reduction of oligodendroglia (i.e., Olig2 + cells), compared to age-matched healthy control (Supplementary Fig. 1b, c). A larger decrease was detected in the mouse corpus callosum (CC), with KO OPCs being about 25% of WT cells (Fig. 2a, c). In contrast, Cit-k KO OPCs density in the preoptic area (POA) was only reduced by one-third, compared to WT (Fig. 2a, e).

A well-defined dorso-ventral gradient of OPC reduction appeared over time in the Cit-k KO forebrain, with the dorsal cortex being the most affected site at P14 (OPC density declined to about 10% of controls, Fig. 2a, b), and striatal/hypothalamic areas maintaining the same density of earlier ages (about 50% of control values in the striatum; about 70% of control values in the POA; Fig. 2a, d, e). Similar results were obtained by counting cells labeled by other OPC markers, such as PDGFRα (412.33 ± 36.11 WT vs. 56.88 KO cells/mm$^2$ in the dorsal cortex; 480.57 ± 17.89 WT vs. 312.34 ± 27.93 KO cells/mm$^2$ in the POA).

The observed dorso-ventral gradient of OPC density was not explained by a different postnatal expansion of dorsal vs. ventral regions containing a stable OPC number in Cit-K KO mouse brain (ANCOVA analysis, Supplementary Table 1). Moreover, it was not due to the failure of dorsal oligodendrogenesis[14,17], as evaluated by three distinct experimental approaches. First, a

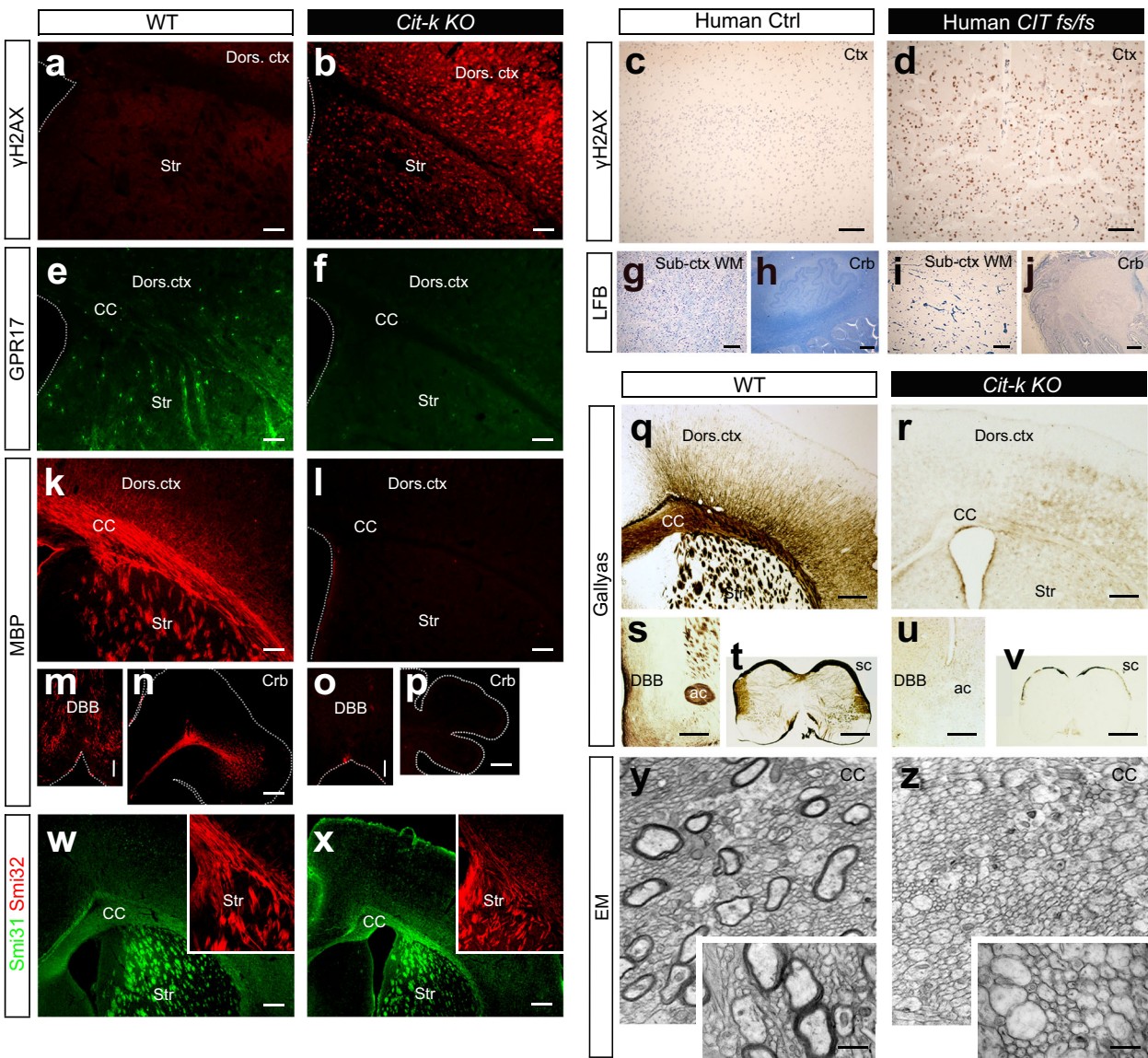

**Fig. 1 DNA damage and defective oligodendroglia differentiation in *Cit-k KO* mouse and *CIT-K fs/fs* human forebrain. a–d** γH2AX+ cell nuclei (red in **a, b**; brown in **c, d**) in P14 *Cit-k KO* mouse (**b**) and human newborn (P1) *CIT-K fs/fs* (**d**) forebrain parenchyma. **e–f** GPR17 (green) protein expression in P14 WT (**e**) and *Cit-k KO* (**f**) mouse brain. **g–j** Myelination (Luxol Fast Blue staining) in the CC and cerebellar white matter of human Ctrl (**g, h**) and *CIT-K fs/fs* subjects (**i, j**). **k–p** MBP (red) protein expression in P14 WT (**k, m, n**) and *Cit-k KO* (**l, o, p**) mouse brain. **q–v** Absence of myelin in *Cit-k KO* mouse brain, as revealed by Gallyas staining (brown) in P14 WT (**q, s, t**) and *Cit-k KO* (**r, u, v**) forebrain and spinal cord. **w, x** Smi32 (red) and Smi31 (green) neurofilament expression pattern reveals the presence of axons in both P14 WT (**w**) and *Cit-k KO* (**x**) mouse brain. **y–z** Representative cross-sectional images of P14 WT (**y**) and *Cit-k KO* (**z**) corpus callosum obtained by EM. Scale bars: 100 μm in **a–p**, 200 μm in **w, x, q–v**, 0.5 μm in **y, z**. ac anterior commissure, CC corpus callosum, DBB diagonal band of Broca, Dors. Ctx. dorsal cortex, EM electron microscopy, P postnatal day, Str striatum, Sub-Ctx WM subcortical white matter, WT wild-type.

retrovirus carrying a GFP reporter (RV-GFP) was injected into the lateral ventricles of P0-P1 WT and *Cit-k KO* mice for fate mapping of dividing precursors in the dorsal SVZ (dSVZ, giving rise to dOPCs in the postnatal phase;[39] Fig. 3a, b left, c left) and of their parenchymal NG2+ dOPC derivatives (Fig. 3b right, c right). At P7, the vast majority of these latter were located in the dorsal cortex and CC (95.8 ± 4.2% WT; 96.9 ± 3.1% *Cit-k KO*). Although the absolute number of GFP+ dSVZ precursors was decreased in *Cit-k KO* compared to WT, no significant change in the number of their parenchymal NG2+ dOPC derivatives was observed (Fig. 3d). An increased ratio of parenchymal GFP+NG2+ dOPCs to their dSVZ ancestors was observed in *KO* mice, while not statistically significant between the two genotypes

(Fig. 3e). These results were corroborated by a second approach using a cell-permeable Cre recombinase Tat-Cre to label dSVZ precursors upon injection into the lateral ventricles of P0 *Cit-k KO*/WT;R26R^YFP mice[40] (Fig. 3f–h). Consistent with former results, at P10, although the absolute number of YFP+ dSVZ precursors was lower in *Cit-k KO* compared to WT, the number and distribution of YFP+ dOPCs (Fig.3g right, h right) did not differ in the two genotypes (Fig. 3i), with about 90% of parenchymal YFP+/PDGFRα+ OPCs residing in the dorsal cortex/CC of both WT and *Cit-k KO* mice. The ratio between the number of parenchymal YFP+ dOPCs and the number of their dSVZ ancestors was higher in *Cit-k KO* compared to WT (Fig. 3j), suggesting a bias of dSVZ precursors toward an

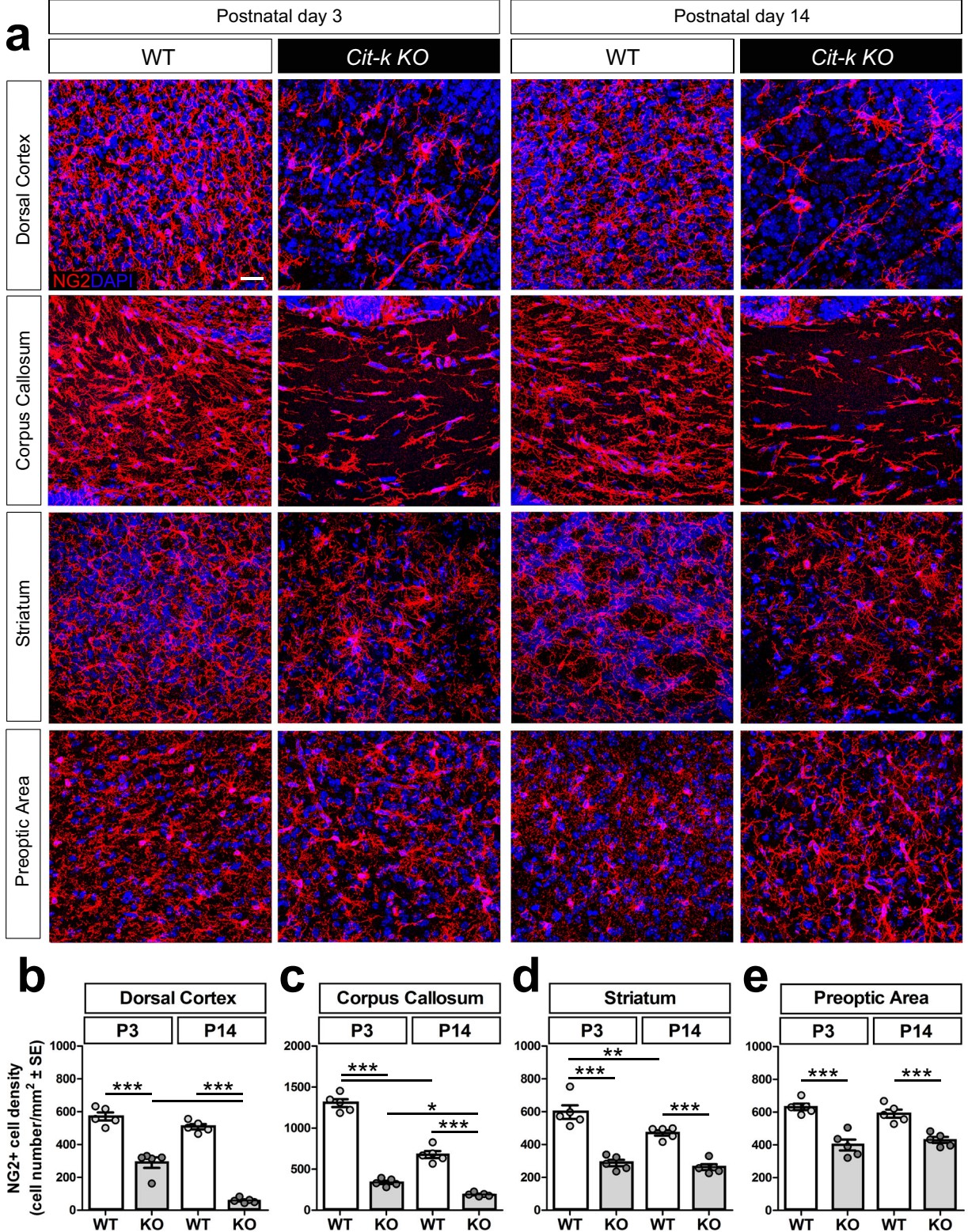

**Fig. 2 *Cit-k* loss differentially affects dorsal and ventral OPCs in the postnatal mouse forebrain. a** NG2+ (red) cells in distinct regions of WT vs. *Cit-k KO* mouse forebrain. DAPI (blue) counterstains cell nuclei. Scale bar: 20 μm. **b–e** NG2+ cell density in the dorsal cortex (**b**), corpus callosum (**c**), striatum (**d**), and preoptica area (**e**) at P3 and P14 (*n* = 5 each; Two-way Anova followed by Bonferroni's Multiple Comparison Test, Dors. Ctx: Genotype effect: $P < 0.0001$; Time effect: $P < 0.0001$; Genotype × Time: $P = 0.0013$. CC: Genotype effect: $P < 0.0001$; Time effect: $P < 0.0001$; Genotype × Time: $P < 0.0001$. Str: Genotype effect: $P < 0.0001$; Time effect: $P = 0.0096$; Genotype × Time: n.s. POA: Genotype effect: $P < 0.0001$; Time effect: n.s.; Genotype × Time: n.s.). Data are mean ± SE. CC corpus callosum, Dors. Ctx. dorsal cortex POA, preoptica area, P postnatal day, WT wild-type. ***$P < 0.001$; **$P < 0.01$; *$P < 0.05$. Source data are provided as a Source Data file.

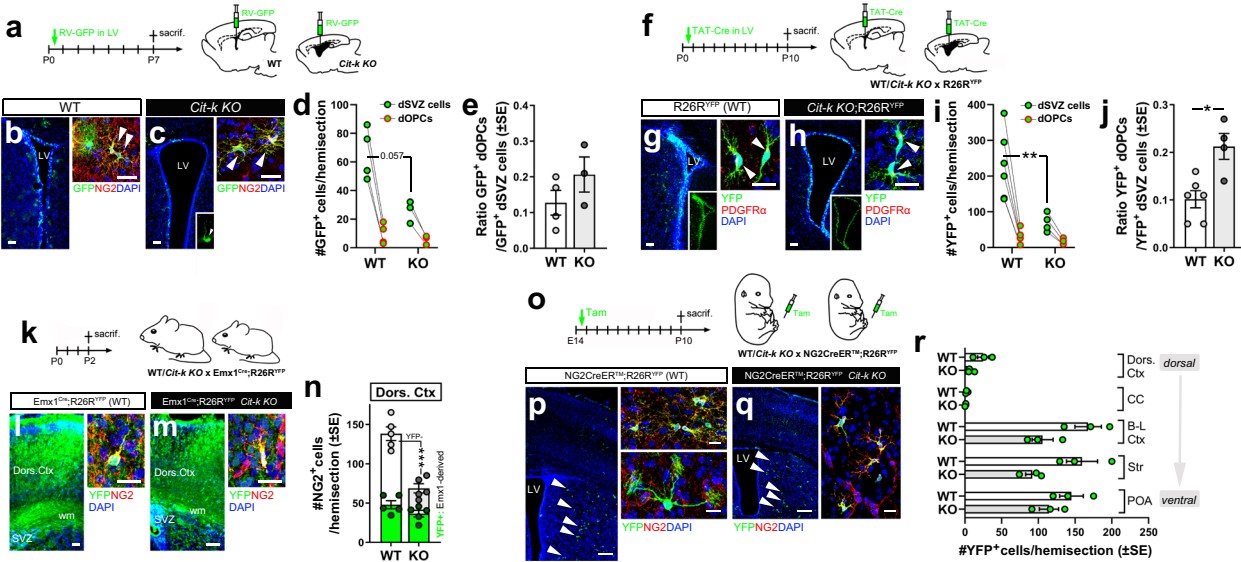

**Fig. 3 Lineage-tracing approaches show that dorsal oligodendrogenesis is preserved in *Cit-k KO* mouse forebrain. a** RV-based lineage-tracing approach. **b**, **c**) Oligodendroglial cells (NG2+, red) in P7 WT (**b** right) or *Cit-k KO* (**c** right) mouse dorsal cortices derived from VZ/SVZ precursors (GFP+, green) targeted by the RV (**b** left, **c** left). White arrowheads in **b** right and **c** right indicate NG2+ OPCs. **d** Absolute numbers/hemisection of RV-targeted (GFP+) cells in the dorsal SVZ (dSVZ; WT n = 4, KO n = 3; two-tailed Mann–Whitney U-test, P = 0.057) and of their OPC (GFP+/NG2+) progeny in the dorsal cortex (dOPCs; WT n = 4, KO n = 3; two-tailed Mann–Whitney U-test, n.s.). **e** Ratio between the number of GFP+/NG2+ cells in the dorsal cortex (dOPCs) and the number GFP+ dSVZ precursors/hemisection (WT n = 4, KO n = 3; two-tailed Mann–Whitney U-test, n.s.). **f** TAT-Cre-based lineage-tracing approach. (**g** left, **h** left) Recombined (YFP+, green) VZ/SVZ precursors in WT (R26R^YFP; **g** left) and *Cit-k KO* (*Cit-k KO*;R26R^YFP; **h** left) mice. (**g** right, **h** right) OPC (PDGFRα+, red) progeny (YFP+, green) in the dorsal cortex of WT (**g** right) and *Cit-k KO* (**h** right) mice. White arrowheads in **g** right and **h** right indicate PDGFRα+ OPCs. **i** Absolute numbers/hemisection of recombined (YFP+) cells in the dorsal SVZ (dSVZ; WT n = 6, KO n = 4; two-tailed Mann–Whitney U-test, P = 0.0095) and of their OPC (YFP+/ PDGFRα+) progeny in the dorsal cortex (dOPCs, WT n = 6, KO n = 4; two-tailed Mann–Whitney U-test, n.s.). **j** Ratio between the number of YFP+/NG2+ cells in the dorsal cortex (dOPCs) and the number YFP+ dSVZ progenitors/ hemisection (WT n = 6, KO n = 4; two-tailed Mann–Whitney U-test, P = 0.0143). **k** Emx1^Cre;R26R^YFP mouse-based lineage-tracing approach. **l**, **m** Progeny (YFP+, green) of Emx1-expressing progenitors in the dorsal cortex and subcortical white matter of WT (Emx1^Cre;R26R^YFP; **l** left) and *Cit-k KO* (Emx1^Cre;R26R^YFP *Cit-k KO*; **m** left) mice. (**l** right, **m** right) Oligodendroglial (NG2+, red) progeny (YFP+, green) in the dorsal cortex of WT (**l** right) and *Cit-k KO* (**m** right) mice. **n** Quantification of NG2+ cells positive or negative for YFP in the dorsal cortex (n = 5 each; two-tailed Unpaired t-test, YFP+: n.s, t(8) = 1.732; YFP-negative: P < 0.0001, t(8) = 9.878). **o** NG2CreER^TM;R26R^YFP mouse-based lineage-tracing approach. **p**, **q** Progeny (YFP+, green) of NG2-expressing progenitors targeted at E14 in the striatum of P10 WT (NG2CreER^TM;R26R^YFP; **p**) and *Cit-k KO* (NG2-CreER^TM;R26R^YFP *Cit-k KO*; **q**) mice (NG2 in red). White arrowheads in **p** and **q** indicate YFP+ cells in the striatum. **r** Mean absolute number/hemisection of YFP+/NG2+ cells in distinct forebrain regions at P10 in WT and *KO* (n = 3 each, two-tailed Mann–Whitney U-test, n.s. in all regions). In **e**, **j**, **n**, **r**) data are mean ± SE. In **d**, **i** lines connect paired samples (i.e., dSVZ cell and dOPC values of the same mouse). DAPI (blue) in **b**, **c**, **g**, **h**, **l**, **m**, **p**, **q** counterstains cell nuclei. Scale bars: 20 μm in **b** right, **c** right, **g** right, **h** right, **l** right, **m** right; 100 μm in **b** left, **c** left, **g** left, **h** left, **l** left, **m** left, 200 μm in **p** left, **q** left, 10 μm in **p** right, **q** right. B-L Ctx baso-lateral cortex, CC corpus callosum, Dors. Ctx. dorsal cortex, dSVZ dorsal subventricular zone, GFP green fluorescent protein, LV lateral ventricle, P postnatal day, POA preoptica area, RV retrovirus, sacrif. sacrifice, Str striatum, WT wild-type, wm white matter, YFP yellow fluorescent protein. ***P < 0.001; **P < 0.01, *P < 0.05. Source data are provided as a Source Data file.

oligodendrogenic fate and/or a higher local expansion of dOPCs in *Cit-k KO* mutants. Finally, we monitored the entire population of dorsally derived OPCs by crossing WT/*Cit-K KO* with Emx1^Cre;R26R^YFP mice (Fig. 3k–m;[14,17]). Again, the absolute number of cortical YFP+ OPCs (Fig.3l right, m right) did not differ between genotypes (Fig. 3n). Together, these results support that dorsal oligodendrogenesis is not negatively affected by the loss of *Cit-k*.

The presence of substantial numbers of OPCs at ventral sites suggested that vOPC genesis was also largely preserved in the *Cit-k KO*. In line with this, at E14, the number of NG2+ cells distributed along the dorso-ventral axis of *Cit-k KO* mouse brain was similar to that of WT samples (Supplementary Fig. 2a–g). Moreover, fate mapping of the progeny of NG2+ cells tagged at E14 in NG2CreER^TM;R26R^YFP[41] × WT/ *Cit-k KO* mice (Fig. 3o–q) confirmed a comparable OPC expansion in both *Cit-k KO* and WT POA (Fig. 3r). In more dorsal parts of the *Cit-k KO* forebrain (i.e., baso-lateral cortex, striatum, and dorsal cortex) the number of YFP+ OPCs displayed a trend for a decrease compared to WT (Fig. 3r), suggesting a local reduction of OPC

expansion/survival or a defective OPC migration to these areas. Defective colonization of the dorsal forebrain by early-born OPCs[14,27] was also supported by the reduced number of YFP-negative OPCs in P2 Emx1^Cre;R26R^YFP × *Cit-k KO* mice compared to WT (Fig. 3n). Thus, *Cit-k* deletion impacts the ability of embryonically generated OPCs to populate the dorsal forebrain and hampers compensation for dorsally derived oligodendroglia loss.

Altogether, these data show that in *Cit-k KO* mouse forebrain: (i) normal (or even relatively higher) numbers of OPCs are generated postnatally in the dorsal cortex but undergo a progressive depletion; (ii) embryonically generated OPCs are produced and persist in the ventral telencephalon, while their colonization of the dorsal forebrain is hampered.

**Cit-k KO dorsal OPCs undergo apoptosis, while Cit-k KO ventral OPCs enter cell senescence.** The preferential decline of *Cit-k KO* dOPCs vs. vOPCs may reflect a differential expression of CIT-K in the two cell subsets. However, equal levels of *Cit-k* mRNA and CIT-K protein were detected in WT forebrain dOPCs

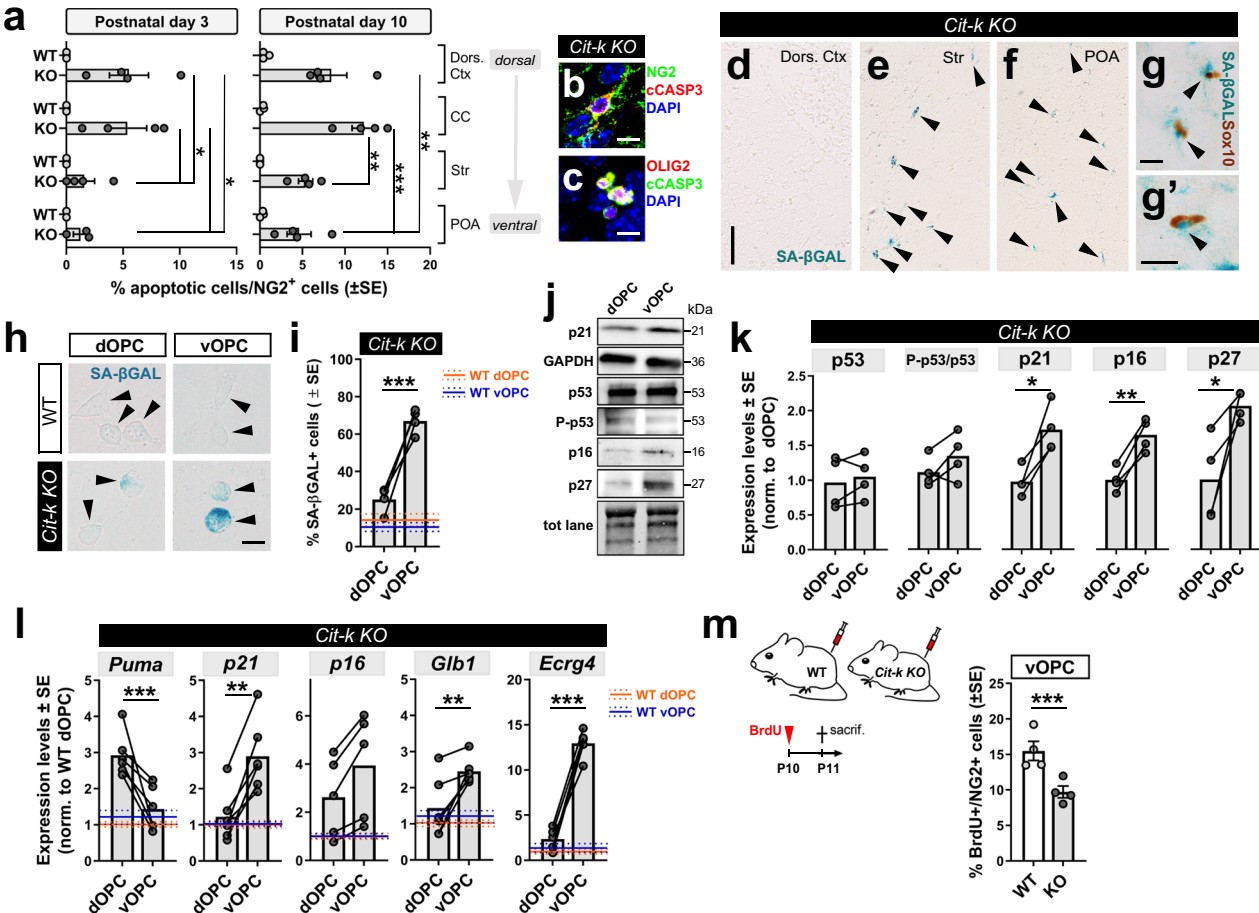

**Fig. 4 Dorsal OPCs undergo apoptosis, while ventral OPCs enter cell senescence in *Cit-k* KO mice. a** Percentage of cCASP3$^+$ OPCs in distinct regions of P3 and P10 WT and *Cit-k* KO mice (Chi-Square Test: P3, $P = 0.0156$ $\chi^2(3) = 10.38$, WT = 3; KO = 4; P10, $P < 0.0001$ $\chi^2(3) = 23.20$, WT = 3; KO = 4). **b**, **c** Apoptotic (cCASP3$^+$, red in **b**, green in **c**) oligodendroglial (NG2$^+$, green in **b**; OLIG2$^+$ red in **c**) cells in *Cit-k* KO mouse brain. DAPI (blue) counterstains cell nuclei. **d–f** SA-βGAL staining (blue) in the dorsal cortex (**d**), striatum (**e**), and preoptica area (**f**) of P14 *Cit-k* KO mice. **g, g′** SA-βGAL$^+$ (blue)/Sox10$^+$ (brown) oligodendroglial cells in *Cit-k* KO mouse striatum. Black arrowheads indicate SA-βGAL$^+$ cells. **h** SA-βGAL staining (blue) in OPCs MACSorted from P10 WT and *Cit-k* KO mice. dOPCs were isolated from the dorsal cortex. vOPCs were isolated from the ventral forebrain. **i** Percentage of SA-βGAL$^+$ cells among MACSorted dOPCs and vOPCs of P10 WT and *Cit-k* KO mice (Chi-Square Test, $P < 0.0001$, $\chi^2(3) = 456.9$, WT dOPCs = 3; WT vOPCs, KO dOPCs and vOPCs = 4 mice). **j, k** Western blotting (**j**) and densitometric analysis (**k**) of p21, p53, phospho-p53 (P-p53), p16, p27 protein expression in P10 *Cit-k* KO dOPCs vs. vOPCs ($n = 4$/group; p21: $P = 0.0249$, t(3) = 4.185; p53: n.s., t(3) = 0.947; phospho-p53/p53: n.s., t(3) = 1.695; p16: $P = 0.0041$, t(3) = 8.017; p27: $P = 0.0449$, t(3) = 3.325, Two-tailed Paired t-test). **l** Expression of mRNAs coding for apoptosis- and senescence-associated proteins in dOPCs and vOPCs MACSorted from P10 WT and *Cit-k* KO mice (Two-way Anova followed by Bonferroni's Multiple Comparison Test: n, P and F values in Supplementary Table 1). **m** Percentage of BrdU$^+$/NG2$^+$ cells 1 days after BrdU administration, in P10 WT and *Cit-k* KO mouse striatum ($n = 4$/group. Chi-Square Test, $P = 0.0007$, $\chi^2(1) = 11.60$). In **a**, **m** data are mean ± SE. In **i**, **k**, **l** lines connect paired samples (i.e., dOPC and vOPC samples from the same mouse). In **i**, **l** orange and blue lines represent mean (solid line) ± SE (dotted lines) of WT dOPCs and WT vOPCs, respectively. Scale bars: 10 μm in **b**, **c**, **g**, **g′**; 100 μm in **d–f**; 5 μm in **h**. BrdU 5-bromo-2′-deoxyuridine, CC corpus callosum, cCASP3 cleaved-caspase 3, dOPC dorsal oligodendrocyte progenitor cell, Dors. Ctx. dorsal cortex, P postnatal day, POA preoptica area, SA-βGAL senescence-associated β-galactosidase, Str striatum, vOPC ventral oligodendrocyte progenitor cell, WT wild-type; *$P < 0.05$; **$P < 0.01$; ***$P < 0.001$. Source data are provided as a Source Data file.

(i.e., isolated from the dorsal cortex and CC) and vOPCs (i.e., isolated from subcortical regions, including septum, striatum, thalamus, and hypothalamus; Supplementary Fig. 3a, b) argued against this interpretation. We therefore suspected a distinct sensitivity of dOPCs vs. vOPCs to *Cit-k* abrogation, and hypothesized that dOPC were prone to cell death, as it was shown for neuronal progenitor cells and cortical neuron subtypes in *Cit-k* null brain[34,42,43]. OPC apoptosis (i.e., positivity for the apoptotic marker cleaved-caspase 3; cCASP3; Fig. 4a–c) in *Cit-K* KO mice was evident at both P3 and P10. Dorsal regions (i.e., dorsal cortex and CC) showed significantly more apoptotic OPCs in comparison to ventral regions (Fig. 4a). Consistently, *Cit-k* KO dOPCs showed higher levels of the pro-apoptotic factor *Puma* (p53-

upregulated modulator of apoptosis, also called *Bbc3*) compared to WT OPCs and to *Cit-K* KO vOPCs (Fig. 4l).

We used multinucleation as a proxy for cytokinesis defects to determine if dOPC apoptosis was linked to incomplete cytokinesis, which was also associated with *Cit-k* loss[33,42,43]. Multinucleated OPCs were observed in both dorsal and ventral areas of P10 *Cit-K* KO mouse forebrain (Supplementary Fig. 3c, d). The fraction of multinucleated OPCs differed at distinct sites, with POA cells showing the lowest rate of multinucleation (Supplementary Fig. 3d). However, these differences showed no correlation with the apoptotic fractions observed in distinct areas (e.g., CC and striatal OPCs showed similar percentages of multinucleated cells, but significantly different apoptotic

fractions; Simple Linear Regression, $R^2 = 0.1981$, $P = 0.1557$, the slope is not significantly different to 0; Supplementary Fig. 3e). In light of the observation that cCASP3 colocalized with both single and multinucleated dOPCs (Fig. 4b, c) and that OPCs were almost completely depleted in P14 *Cit-K KO* dorsal cortex - where multinucleated cells accounted only for a third of the entire OPC population at P10 - these findings suggest that cell death in *Cit-k KO* OPCs is not a direct consequence of incomplete cytokinesis.

Thus, both *Cit-k KO* dOPCs and vOPCs were generated, but dOPCs underwent apoptosis with higher frequency compared to vOPCs. Since apoptosis and senescence are linked and often alternative cell responses to stressors[44], and senescence was reported as a direct consequence of *CIT-K* loss in medulloblastoma[45,46], we evaluated features of senescence in OPCs of mutant vs. control tissue. We found numerous Sox10[+] oligodendroglial cells showing senescence-associated β-galactosidase (SA-βGAL) activity in the striatum and POA of *Cit-k KO* P10 mice (Fig. 4e–g'). In contrast, SA-βGAL[+] senescent cells were not detected in the *Cit-k KO* dorsal cortex and CC (Fig. 4d). Further, the majority of P10 MACS-sorted *Cit-k KO* vOPCs were SA-βGAL[+] (Fig. 4h, i). Consistent with this finding, vOPCs showed increased expression of senescence-associated markers (i.e., p21, p16, p27 proteins and *p21, p16, Glb1,* and *Ecrg4* mRNAs;[47,48]), which was not detected in *KO* dOPCs (Fig. 4j–l). Moreover, *Cit-k KO* vOPCs showed reduced proliferative activity (i.e., incorporation of BrdU) compared to WT vOPCs (Fig.4m), a typical hallmark of senescent cells[48]. Distinct expression or activation of p53 may be upstream to either *Puma* or *p21* upregulation[44]. Yet, similar levels of p53 and of phospho-p53 were found in *Cit-k KO* dOPC and vOPC (Fig. 4j, k), suggesting that different events independent/downstream of p53 may be involved in driving the activation of either the apoptotic or the senescence pathway in the two cell subsets.

Together these data indicate that the loss of *Cit-k* results in alternative postnatal cell fates in dOPCs and vOPCs, i.e., cell death in dOPCs and cell senescence in vOPCs.

**dOPCs and vOPCs exhibit distinct cell-autonomous vulnerabilities to *Cit-k* loss.** To assess whether the alterations of *Cit-k KO* OPCs depend on environmental factors, we grafted telencephalic cells dissociated from WT β-actin-GFP mouse forebrain into the deep dorsal cortex and striatum of *Cit-k KO* mice (Supplementary Fig. 4a). *Cit-k KO* transplant recipients were sacrificed at 6 and 14 days post-transplantation. Integrated cells were very abundant, dispersed throughout the entire forebrain (Supplementary Fig. 4a), and included almost exclusively oligodendroglial cells (i.e., 76.01 ± 5.91% of all GFP + cells; Supplementary Fig. 4b, d). In both the dorsal cortex and striatum/thalamus, the density of grafted oligodendroglia did not decrease over time, but rather progressively expanded (Supplementary Fig. 4c), indicating that the depletion of resident dOPCs depended on cell-intrinsic features, rather than on extrinsic factors affecting OPC survival. Moreover, at difference with endogenous OPCs, in the subcortical white matter and striatum, several grafted cells proceeded along the lineage, displaying pre-myelinating markers, such as GPR17, and alignment of processes to axons (Supplementary Fig. 4d and insets). This latter finding, together with the observed expansion of grafted OPCs in the striatum/thalamus (Supplementary Fig. 4c), suggests that the acquisition of senescent properties in *Cit-k KO* vOPCs also depended on cell-intrinsic factors, rather than on environmental senescence-inducers. Accordingly, in both *Cit-k KO* dorsal and ventral forebrain tissues, the levels of pro-inflammatory factors (i.e., *Il-1β, Tnfα, Nos2, CxCl1*), that could contribute to both dOPCs cell death and vOPC senescence[44], were not increased compared to WT (Supplementary Fig. 4e).

The role of cell-intrinsic factors in the alternative fates of *Cit-k KO* dOPCs vs. vOPCs was instead addressed in *Cit-k* conditional mutants. A dorso-ventral gradient of OPC abundance — resembling that observed in the germinal *Cit-k KO* — was detected in P14 Sox10[Cre];Cit-k[fl/fl] mouse forebrain, where *Cit-k* was deleted only in oligodendroglia (Fig. 5a, b). Similar to *Cit-k KO* dOPCs, Sox10[Cre];Cit-k[fl/fl] dOPCs showed higher apoptotic fractions (Fig. 5c) and a slight upregulation of the pro-apoptotic factor *Puma*, compared to their ventral counterparts (Fig. 5e). On the other side, similar to what was found in the germinal *Cit-k KO*, Sox10[+]/SA-βGAL[+] oligodendroglial cells were detected in the ventral Sox10[Cre];Cit-k[fl/fl] forebrain (Fig. 5d) and Sox10[Cre];Cit-k[fl/fl] vOPCs showed higher levels of senescence markers compared to dOPCs (i.e., *p21* and *Ecrg4* mRNAs, Fig. 5e). Expression of the myelin-associated protein MBP was also perturbed in P14 Sox10[Cre];Cit-k[fl/fl] forebrain. The dorsal cortex and CC exhibited the greatest reduction, but also ventral areas (e.g., periventricular striatum, medial septum, lateral cortex, and POA) showed reduced anti-MBP immunolabeling in Sox10[Cre];Cit-k[fl/fl] mice compared to Sox10[Cre] controls (Supplementary. Fig. 5a, b). These data are in line with a cell-intrinsic role of *Cit-k* loss in producing divergent dOPCs vs. vOPC alterations.

To assess whether the alternative fates of *Cit-k KO* dOPCs vs. vOPCs where linked to their distinct developmental origin, we generated two conditional mouse lines to selectively delete *Cit-k* in either dorsal VZ/SVZ (Emx1[Cre];R26R[YFP];Cit-k[fl/fl]) or ventral MGE/AEP/POA (Nkx2.1[Cre];R26R[YFP];Cit-k[fl/fl]) precursors and in their parenchymal derivatives (Fig. 5f, g). The density of dorsally derived YFP[+] NG2[+] OPCs was significantly reduced (Fig. 5h) and higher fractions of apoptotic OPCs were detected (Fig. 5i) in P14-P16 Emx1[Cre];R26R[YFP];Cit-k[fl/fl] mouse dorsal cortex and CC, compared to Emx1[Cre];R26R[YFP] controls. However, in both regions the overall NG2[+] OPC density did not vary between the two genotypes (Fig. 5h), indicating a compensatory expansion of ventrally derived WT OPCs at dorsal sites. In line with this interpretation, Emx1[Cre];R26R[YFP];Cit-k[fl/fl] mouse dorsal cortex exhibited a relatively preserved MBP expression pattern (at least in the deeper layers; Supplementary Fig. 5c, d), although showing a 30% reduction of CC1[+] mature oligodendrocytes (Supplementary Fig. 5j, k). However, cortical size was decreased (Supplementary Fig. 5c, d, g) and BLBP[+] astrocyte reduced in number (Supplementary Fig. 5h, i), consistent with a global defect in dorsal VZ/SVZ derivatives in Emx1[Cre];R26R[YFP];Cit-k[fl/fl] mice.

In contrast, the density of ventrally derived YFP[+] NG2[+] OPCs did not differ in P14-P16 Nkx2.1[Cre];R26R[YFP];Cit-k[fl/fl] vs Nkx2.1[Cre];R26R[YFP] control POA (the only region in which YFP[+] OPCs could be detected at this stage; Fig. 5j) and MBP expression pattern did not show any apparent alteration (Supplementary Fig. 5e, f). Together these data indicate that cell-intrinsic features associated with OPC developmental heterogeneity underlie the alternative cell fates of forebrain *Cit-k KO* dOPCs and vOPCs.

**Cit-k KO dorsal and ventral OPCs exhibit DNA damage and display a differential response to oxidative stress.** The related but distinct fates of apoptosis and senescence may point to a differential accumulation or response to DNA damage in dOPCs vs. vOPCs[44]. To assess DNA damage, we quantified γH2AX in acutely isolated OPCs of P10 WT vs. *Cit-k KO* mice. While WT OPCs hardly showed any positive γH2AX labeling (Fig.6a), 90% of both *Cit-k KO* dOPCs and vOPCs exhibited γH2AX foci (Fig. 6a, b). *Cit-k KO* dOPCs and vOPCs displayed the same number of nuclear γH2AX[+] foci (Fig. 6b) and similar levels of nuclear γH2AX staining integrated density (Supplementary Fig. 6a), indicating an equivalent amount of DNA lesions. These results were confirmed by Western Blot analysis of γH2AX

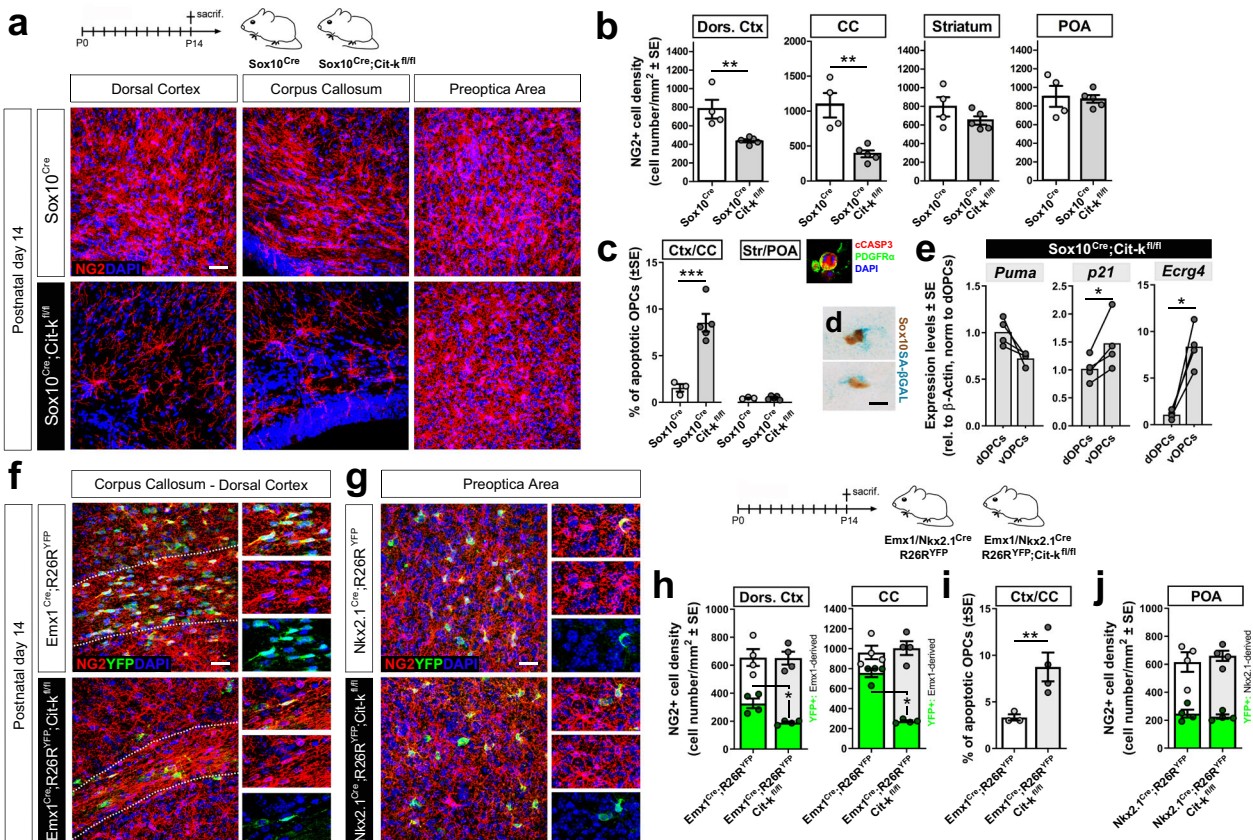

**Fig. 5 Alternative fates of *Cit-k KO* dorsal vs. ventral OPCs are due to cell-intrinsic factors and associated with their diverse developmental origin.** **a** NG2+ (red) cells in distinct regions of Sox10[Cre] (Ctrl) vs. Sox10[Cre];Cit-k[fl/fl] mouse forebrain at P14. **b** NG2+ cell density in the dorsal cortex, corpus callosum, striatum and preoptica area at P14 (Sox10[Cre] $n = 4$; Sox10[Cre];Cit-k[fl/fl] $n = 5$. Two-tailed Unpaired $t$-test, Dors. Ctx: $P = 0.0073$, t(7) = 3.741; CC: $P = 0.0038$, t(7) = 4.249; Str: n.s.; POA: n.s.). **c** Percentage of cCASP3+ OPCs in distinct regions of P14 Sox10[Cre] (Ctrl; $n = 3$) vs. Sox10[Cre];Cit-k[fl/fl] ($n = 5$) mice (Chi-Square Test, Dors.Ctx/CC: $P < 0.0001$, $\chi^2(1) = 15.98$). (inset) Representative PDGFRα+(green)/cCASP3+(red) OPC in Sox10[Cre];Cit-k[fl/fl] mouse cortex. **d** SA-βGAL+(blue)/Sox10+(brown) oligodendroglial cells in Sox10[Cre];Cit-k[fl/fl] mouse striatum. **e** Expression of mRNAs coding for apoptosis- and senescence- associated proteins in dOPCs vs. vOPCs MACSorted from P10 Sox10[Cre];Cit-k[fl/fl] mice (Two-tailed Paired $t$-test, $n = 4$/group; *Bbc3*, n.s.; *p21*, $P = 0.049$, t(3) = 3.204; *Ecrg4*: $P = 0.011$, t(3) = 5.744). **f** NG2+ (red) cells in the CC and deep layers of the dorsal cortex of Emx1[Cre];R26R[YFP] (Ctrl) vs. Emx1[Cre];R26R[YFP];Cit-k[fl/fl] P14 mice. YFP (green)-positivity identifies cells derived from dorsal Emx1+ progenitors. **g** NG2+ (red) cells in the preoptic area of Nkx2.1[Cre];R26R[YFP] (Ctrl) vs. Nkx2.1[Cre];R26R[YFP];Cit-k[fl/fl] P14 mice. YFP (green)-positivity identifies cells derived from ventral Nkx2.1+ progenitors. **h** NG2+ and NG2+/YFP+ cell density in the dorsal cortex and corpus callosum of P14-P16 Emx1[Cre];R26R[YFP] (Ctrl, $n = 4$) vs. Emx1[Cre];R26R[YFP];Cit-k[fl/fl] ($n = 4$. Two-tailed Mann–Whitney U-test, Dors. Ctx: tot: n.s.; YFP + : 0.028; CC: tot: n.s.; YFP + : 0.028). **i** Percentage of cCASP3+ OPCs in Dors.Ctx/CC of P14-P16 Emx1[Cre];R26R[YFP] (Ctrl, $n = 3$) vs. Emx1[Cre];R26R[YFP];Cit-k[fl/fl] ($n = 4$) mice (Chi-Square Test: Dors.Ctx/CC: $P = 0.0038$, $\chi^2(1) = 8.362$). **j** NG2+ and NG2+/YFP+ cell density in the preoptica area of P14-P16 Nkx2.1[Cre];R26R[YFP] (Ctrl, $n = 4$) vs. Nkx2.1[Cre];R26R[YFP];Cit-k[fl/fl] ($n = 4$. Two-tailed Mann–Whitney U-test, tot: n.s.; YFP + : n.s.). Data in **b**, **c**, **h**, **i**, **j** are mean ± SE. In **e** lines connect paired samples (i.e., dOPC and vOPC samples from the same mouse). DAPI (blue) in **a**, **f**, **g** counterstains cell nuclei. Scale bars: 30 μm in **a**, **f**, **g**; 10 μm in **d**. CC corpus callosum, Ctrl control, Dors. Ctx. dorsal cortex, POA preoptica area, P postnatal day. ***$P < 0.001$; **$P < 0.01$, *$P < 0.05$. Source data are provided as a Source Data file.

expression (Fig. 6f, g) and by quantification of *Bach2* mRNA, whose transcription is suppressed in response to DNA damage[49] (Supplementary Fig. 6b). A similar accumulation of γH2AX+ foci and nuclear γH2AX+ staining was found in Sox10[Cre];Cit-k[fl/fl] dOPCs and vOPCs (Supplementary Fig. 6c), which also showed an equivalent suppression of *Cit* and *Bach2* expression (Supplementary Fig, 6d). γH2AX+ DNA lesions were also detected in the dSVZ and CC of P14 Emx1[Cre];R26R[YFP];Cit-k[fl/fl] (Supplementary Fig, 6e) and in the vSVZ and POA of P14 Nkx2.1[Cre];R26R[YFP];Cit-k[fl/fl] mice (Supplementary Fig, 6f). This shows that, upon *Cit-K* deletion, dOPCs and vOPCs are exposed to the same primary damage.

DNA damage responses are known to induce the production of reactive oxygen species (ROS)[50–52], which were determined by dihydroethidium (DHE) fluorescence[53] (Fig. 6c, d) and by the colorimetric Nitroblue Tetrazolium (NBT) reduction assay[54]

(Fig. 6e). These analyses revealed higher ROS in *Cit-k KO* dOPCs, as compared to vOPCs (Fig. 6c–e), suggesting that different responses to oxidative stress may account for the selective loss of dOPCs and senescence in vOPCs. In line with this idea, although the transcript coding for the NRF2 transcription factor, a master regulator of the anti-oxidant response[55], was significantly upregulated in both *Cit-k KO* OPC subsets (Fig. 6j, Supplementary Fig, 7), NRF2 protein was reduced in *Cit-k KO* dOPCs, compared to vOPCs (Fig. 6f–i). This was not due to a general blockade of protein translation in apoptotic cells, as dOPCs exhibiting low NFR2 expression displayed a non-pyknotic healthy morphology (Fig. 6h). Consistent with a diverse ability of *Cit-k KO* dOPCs vs. vOPCs to set up a NRF2-mediated anti-oxidant response, a panel of NRF2-target genes (i.e., *Sod1*, *Gpx1*, *Hmox1*, and *Nqo1*) were significantly upregulated only in *Cit-k KO* vOPCs, while *Sod2*, *Cat* and *Gpx3* were even paradoxically

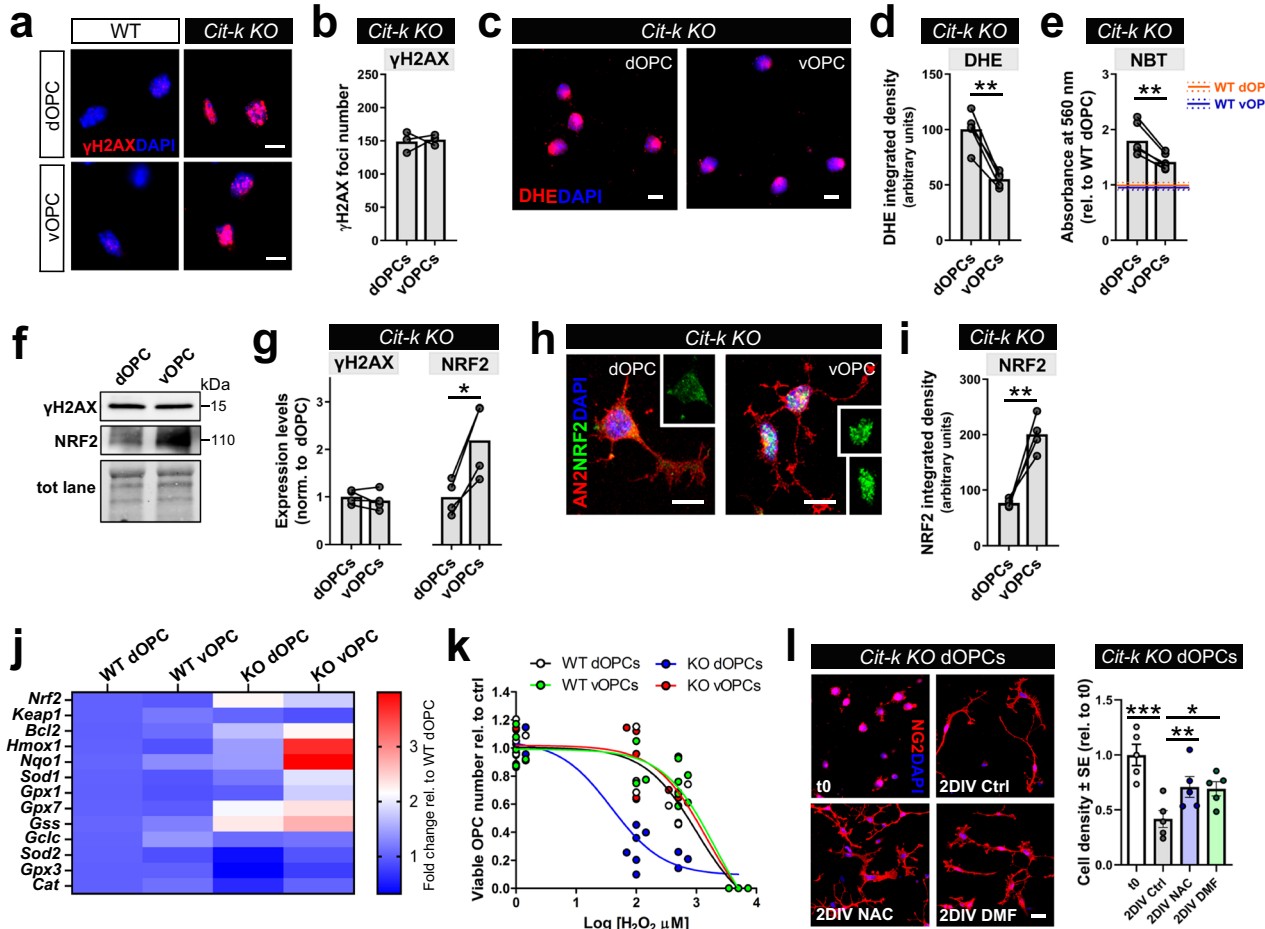

**Fig. 6 *Cit-k KO* dorsal OPCs are more vulnerable to oxidative stress compared to *Cit-k KO* ventral OPCs. a** γH2AX expression in acutely MACS-isolated OPCs of P10 *Cit-k KO* mice. **b** Number of γH2AX+ foci in dOPCs vs. vOPCs (*n* = 3 each, two-tailed Mann–Whitney U-test, n.s.). **c** DHE labeling in acutely MACS-isolated OPCs of P10 *Cit-k KO* mice. **d** DHE integrated density in dOPCs vs. vOPCs (*n* = 5 each, Two-tailed Paired *t*-test, **$P$ = 0.0013, $t(4)$ = 8.043). **e** NBT reduction assay in *Cit-k KO* dOPCs vs. vOPCs (normalized to dOPC mean value. *n* = 4/group, Two-way Anova: Genotype: $P < 0.0001$, $F(1,21)$ = 92.83; Region: $P = 0.0067$, $F(1,21)$ = 9.045; Genotype × Region: $P = 0.0204$, $F(1,21)$ = 6.288). Orange and blue lines represent mean (solid line) ± SE (dotted lines) of WT dOPCs and WT vOPCs, respectively. **f, g** Western blotting (**f**) and densitometric analysis (**g**) of γH2AX and NRF2 protein expression in P10 *Cit-k KO* dOPCs vs. vOPCs (*n* = 4/group; γH2AX: n.s.; NRF2: $P = 0.0155$, $t(3)$ = 4.984, Two-tailed Paired *t*-test). **h** NRF2 (green) expression in MACSorted *Cit-k KO* AN2+ (red) dOPCs and vOPCs in vitro. DAPI (blue) counterstains cell nuclei. **i** NRF2 staining integrated density in MACSorted *Cit-k KO* dOPCs vs. vOPCs (*n* = 4/group; $P = 0.0087$, $t(3)$ = 6.129, Two-tailed Paired *t*-test). **j** Heatmap of the qRT-PCR analysis of the mRNAs of *Nrf2*, *Keap1*, and NRF2-target genes in dOPCs and vOPCs MACSorted from P10 WT and *Cit-k KO* mice (Two-way Anova followed by Bonferroni's Multiple Comparison Test; n, P, and F values in Supplementary Table 1). Dot plots are included in Supplementary Fig 7. **k** Log(inhibitor) vs. response curve (Non-linear regression inhibition curve) representing percentages of viable OPCs after an acute $H_2O_2$ treatment (0, 100, 500, 5000 μM). LC50 *KO* dOPCs = 38.96 μM, $R^2$ = 0.885; LC50 *KO* vOPCs = 1481 μM, $R^2$ = 0.859; LC50 WT dOPCs = 1015 μM, $R^2$ = 0.895; LC50 WT vOPCs = 2046 μM, $R^2$ = 0.907). **l** *Cit-k KO* dOPC density at t0 (i.e., 30 min after plating) and at t = 2 DIV in Ctrl conditions vs. in presence of NAC or DMF (*n* = 5/group; $P < 0.0001$, $F(3)$ = 53.86, One-way Anova Repeated Measures). In **l** data are mean ± SE. In **b, d, e, g, i** lines connect paired samples (i.e., dOPC and vOPC samples from the same mouse). Scale bars: 5 μm in **a, c, h**; 10 μm in **l**. DHE dihydroethidium, DIV days in vitro, DMF dimethyl fumarate, dOPC dorsal oligodendrocyte progenitor cell, $H_2O_2$ hydrogen peroxide, LC50 inhibitory concentration 50—concentration that produces 50% decrease in viable cells, NAC N-acetyl-L-cysteine, NBT Nitroblue Tetrazolium, vOPC ventral oligodendrocyte progenitor cell, WT wild-type; P postnatal day, γH2AX phosphorylated histone H2AX. *$P < 0.05$; **$P < 0.01$; ***$P < 0.001$. Source data are provided as a Source Data file.

downregulated in *Cit-k KO* dOPCs (Fig. 6j; Supplementary Fig. 7).

These findings suggest that *Cit-k KO* dOPCs are uniquely vulnerable to oxidative stress. To test this possibility, we exposed acutely isolated WT and *Cit-k KO* OPCs to increasing concentrations of $H_2O_2$. *Cit-k KO* dOPCs underwent cell death at sublethal $H_2O_2$ concentrations, while *Cit-k KO* vOPCs tolerated $H_2O_2$ concentrations comparable to WT OPCs (Fig. 6k; lethal concentration 50, LC50, *KO* dOPCs = 38.96 μM, *KO* vOPCs = 1481 μM, WT dOPCs = 1015 μM, WT vOPCs = 2046 μM). Consistent with a key role of oxidative stress and NRF2

dysregulation in *Cit-k KO* dOPC death, treatment with either the anti-oxidant agent N-acetylcysteine (NAC) or the NRF2-activator dimethyl fumarate (DMF) resulted in a significant and equivalent increase of *Cit-k KO* dOPC survival in vitro (Fig. 6l).

Similarly, while reducing ROS and DNA damage in *Cit-k KO* dOPCs (Fig. 7b, c), the chronic postnatal in vivo treatment with NAC (Fig. 7a) led to a 2.5-fold increase in dOPC density (Fig. 7d, e) and greatly reduced the cell apoptotic fraction (Fig. 7f) of P10 *Cit-k KO* dOPCs. NAC treatment also reduced the mitotic activity of dOPCs in *Cit-k KO* CC (Fig. 7g), suggesting promotion of differentiation. However, NAC did not restore the expression

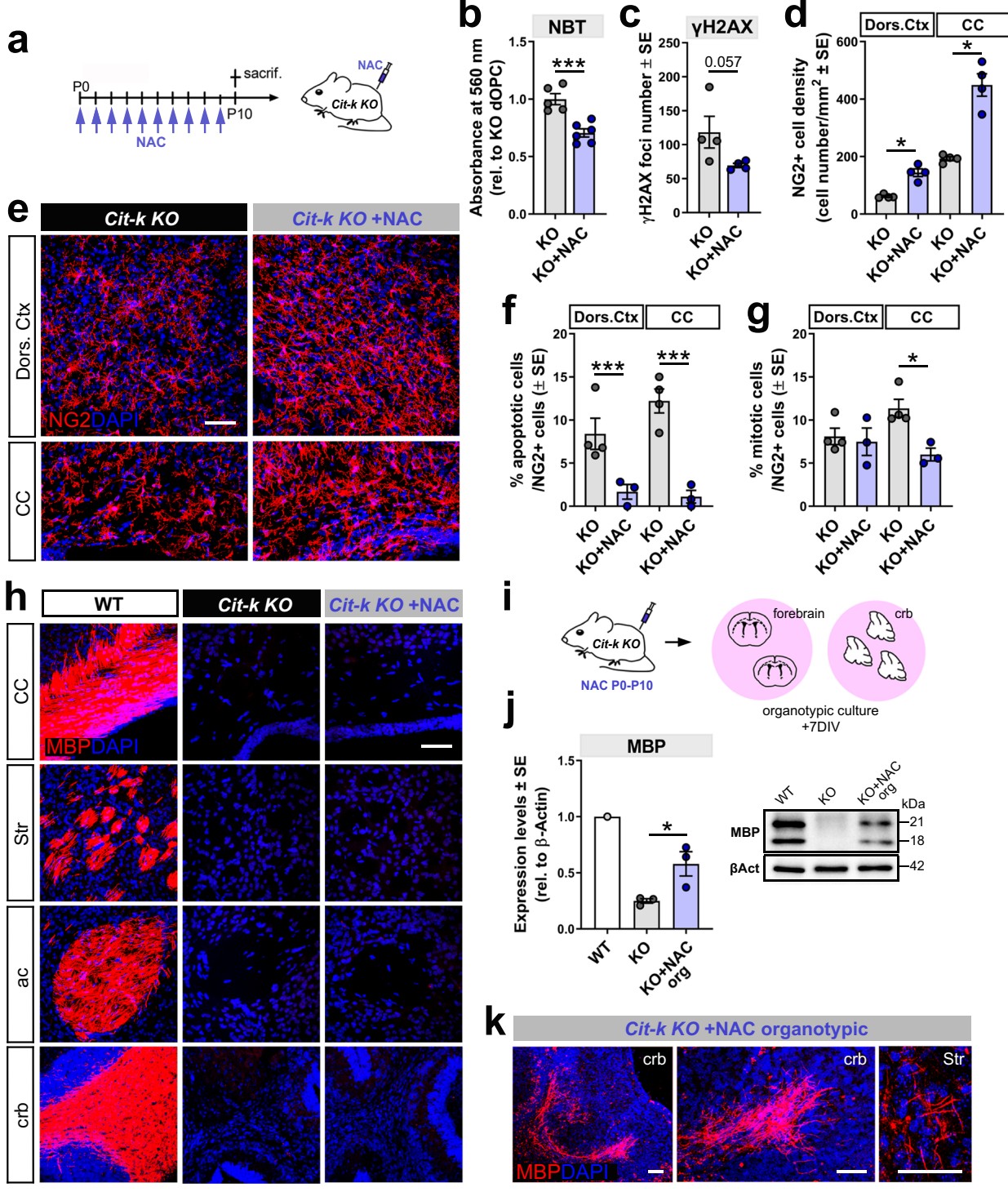

of MBP (Fig.7h) or of other markers of oligodendroglia differentiation (i.e., CC1 or *Gpr17, Plp1,* and *Mbp* mRNAs; Supplementary Fig. 8a, b), neither reduced expression of *p53* and *p21* (Supplementary Fig. 8a) in *Cit-k KO* mouse brain within the first 2 postnatal weeks. Nevertheless, when *Cit-k KO* cells were given more time to differentiate in organotypic slices cultured in a medium supplemented with NAC (Fig. 7i), MBP protein levels increased and MBP⁺ tubular structures were detected (Fig. 7j, k). These data indicate that *Cit-k KO* dOPC vs. vOPC alternative cell fates are associated with a differential ability to set up NRF2-dependent anti-oxidant responses and that oxidative stress and

NRF2 dysregulation are key players in the postnatal depletion of *Cit-k KO* dOPCs.

**WT dorsal and ventral OPCs set up different responses to DNA damage in vitro.** We next wondered whether the specific vulnerability of mutant dOPCs was a feature shared also by WT cells, when challenged by specific stressors. The similar response of WT OPC populations when treated with H₂O₂ (Fig. 6k), indicated that an acute oxidative stress per se may not be able to elicit distinct responses in WT dOPCs vs. vOPCs. Thus,

**Fig. 7 NAC treatment partially rescues the oligodendroglia phenotype in *Cit-k KO* mouse forebrain. a** NAC treatment. **b** NBT reduction assay in dOPCs acutely isolated from P10 *Cit-k KO* ($n = 5$) vs. NAC-treated *Cit-k KO* ($n = 6$) mice (normalized to *Cit-k KO* dOPC mean value. Two-tailed Unpaired *t*-test, $P = 0.0007$, t(9) = 5.006). **c** Number of $\gamma$H2AX$^+$ foci in dOPCs acutely isolated from P10 *Cit-k KO* vs. NAC-treated *Cit-k KO* mice ($n = 4$/group; two-tailed Mann–Whitney U-test, $P = 0.0571$). **d** NG2$^+$ cell density in the dorsal cortex and CC of *Cit-k KO* mice vs. NAC-treated *Cit-k KO* mice ($n = 4$ each, two-tailed Mann–Whitney U-test, $P = 0.0286$ for both dorsal cortex and CC). **e** NG2$^+$ (red) cells in the dorsal cortex and CC of P10 *Cit-k KO* mice vs. *Cit-k KO* mice after NAC treatment. **f** cCASP3$^+$/NG2$^+$ cells in the dorsal cortex and CC of *Cit-k KO* mice ($n = 4$) vs. NAC-treated *Cit-k KO* mice ($n = 3$, Chi-square test, dorsal cortex $\chi^2(1) = 19.05$, CC $\chi^2(1) = 27.26$, $P < 0.0001$ for both). **g** Percentage of mitotic (i.e., PH3$^+$) NG2$^+$ cells in the dorsal cortex and CC of *Cit-k KO* mice ($n = 4$) vs. NAC-treated *Cit-k KO* mice ($n = 3$, Chi square test, dorsal cortex: n.s., $\chi^2(1) = 0.00253$; CC: $P = 0.0323$, $\chi^2(1)$: 4.581). **h** Anti-MBP (red) immunolabeling in P10 WT, *Cit-k KO* and NAC-treated *Cit-k KO* mouse major white matter tracts. **i** Experimental approach to obtain forebrain and cerebellar organotypic cultures from NAC-treated P10 *Cit-k KO* mice. **j** Western blotting analysis of MBP protein expression in P10 WT ($n = 1$), *Cit-k KO* mouse forebrain tissue ($n = 3$) and *Cit-k KO* organotypic cultures ($n = 3$; one-tailed Mann–Whitney U-test; $P = 0.05$). **k** Anti-MBP (red) immunolabeling in *Cit-k KO* cerebellar and forebrain organotypic cultures. DAPI (blue) in **e**, **h**, **k** counterstains cell nuclei. Scale bars: 50 µm. Data are mean ± SE. ac anterior commissure, CC corpus callosum, cCASP3 cleaved-caspase 3, crb cerebellar white matter, DIV days in vitro, Dors. Ctx. dorsal cortex, NAC N-acetyl cisteine, P postnatal day, sacrif. sacrifice, Str striatum, WT wild-type. *$P < 0.05$; ***$P < 0.001$. Source data are provided as a Source Data file.

we hypothesized that DNA damage could be the key stressor that uncovered distinct vulnerabilities in OPCs. To test this hypothesis, OPCs were isolated from P8 WT mouse dorsal cortex and ventral forebrain, cultured for 48 h at high density, and then incubated with titrated concentrations of the cross-linking agent cisplatin (Fig. 8a). 24 h post-treatment with 100 nM cisplatin, the vast majority of both dOPCs and vOPCs were $\gamma$H2AX$^+$, and showed similar levels of DNA damage (Supplementary Fig. 9a, b and Fig. 8h, i). dOPCs showed to be intrinsically more sensitive to cisplatin (Fig. 8b), with a calculated LC50 of 48 nM, compared to 173.7 nM cisplatin of vOPCs (Fig. 8c). In control conditions, the percentage of mitotic cells did not differ in the two cell populations ($4.55 \pm 1.04\%$ dOPCs; $5.74 \pm 0.51\%$ vOPCs; $P = 0.22$ Student's *t*-test), indicating that distinct proliferative rates did not account for the differential behavior of dOPC and vOPCs. To assess whether such a differential response to cisplatin was associated with OPC developmental origin, we isolated OPCs from the entire forebrain of P10 Emx1$^{Cre}$;R26R$^{YFP}$ mice and assessed the abundance of dorsally derived (i.e., YFP$^+$) vs. ventrally derived (YFP-negative) OPCs 48 h after treatment with 100 nM cisplatin (compared with ctrl condition, i.e., 48 h after treatment with vehicle; Fig. 8d). Both OPC subsets decreased significantly after cisplatin treatment (Fig. 8e–g). However, dorsally derived YFP$^+$ OPCs showed higher vulnerability (Fig. 8g). Like *Cit-k KO* vOPCs, 24 h after cisplatin treatment, WT vOPCs showed higher expression of senescence markers, such as p21, p16, and p27, compared to dOPCs (Fig. 8h–i), suggesting the activation of a cellular senescence program in vOPCs. Similar to *Cit-k KO* dOPCs, injured WT dOPCs displayed lower NRF2 expression (Fig. 8h–i), and higher levels of intracellular ROS (Fig. 8j), compared to vOPCs. In line with the idea that WT dOPC vulnerability to DNA damage depended on oxidative stress, NAC co-treatment resulted in a significant and dose-dependent rescue of dOPC numbers after cisplatin treatment (Fig. 8k, l). Taken together, these findings indicate that the differential vulnerability of dOPC vs. vOPC subsets to DNA damage also extends to *Cit-k* independent conditions and reflects inherent OPC developmental diversity.

## Discussion

To shed light on OPC heterogeneity, here we analyzed a mouse model of microcephaly where the loss of *Cit-k* lead to a prominent hypomyelination and to either cell death or senescence in dOPCs and vOPCs, respectively. Experiments in conditional *Cit-k* mutants showed that cell-autonomous factors associated with dorsal vs. ventral developmental origin underlie dOPC vs. vOPC fate divergence. Mechanistically, while not showing a distinct amount of DNA lesions or different levels of p53/phospho-p53 expression, the two *Cit-k KO* OPC subsets displayed a diverse

accumulation of ROS and a different ability to set up a NRF2-mediated anti-oxidant response. Anti-oxidant or NRF2-activating drugs were able to support *Cit-k KO* dOPC survival, pointing to oxidative stress and NRF2 dysregulation as key players in *Cit-k KO* dOPC depletion. Developmental heterogeneity also influenced the response of WT forebrain dOPCs vs. vOPCs to cisplatin-induced DNA damage in vitro. While WT dOPCs displayed a higher ROS-dependent vulnerability, vOPCs were more resilient and expressed higher levels of NRF2 and senescence markers. Thus, diversity in dOPC vs. vOPC responses to DNA damage was not restricted to *Cit-k* loss-associated conditions, and — also in case of WT OPCs — depended on a different cell-intrinsic ability to counteract oxidative stress.

The developing mammalian brain hosts a heterogeneous population of neuronal and glial progenitors, that are differentially affected by developmental disorders such as microcephaly[56–58]. While the microcephaly syndromes are relatively rare, the study of these neurodevelopmental disorders can reveal molecular mechanisms critically involved in progenitor subtype diversity. Reduced neurogenesis and increased apoptosis in the dorsal SVZ/cortex and ganglionic eminences have been reported in the developing *Cit-k null* brain[33,34,42,43]. Although OPCs arise from the same niches, in our study both dorsal and ventral oligodendrogenesis were substantially preserved. In the first 7–10 days of life, dOPC production appeared even increased in *Cit-k KO* compared to WT forebrain, possibly to compensate for the defective early colonization of dorsal regions by vOPCs (similar to what found in Mash1 mutants[59]). At later stages, *Cit-k KO* vOPCs were also unable to counterbalance dOPC loss (even when *Cit-k* was ablated only in oligodendroglia). Although we cannot completely exclude that vOPCs may undergo cell death while moving toward more dorsal areas, the reduced vOPC apoptosis and the absence of vOPC migrating streams (see Fig. 2a, Fig. 5a) suggest migration defects possibly linked to vOPC senescent phenotype[60–63]. Yet, neither migration impairment, nor senescence was detected in immature *Cit-k KO* neurons[33,34]. Thus, OPCs respond differently to *Cit-k* ablation compared to neuronal progenitors and neuroblasts. This may reflect a cell type-specific requirement for CIT-K in distinct cell processes and/or the interaction between the effects of *Cit-k* ablation and diverse cell-intrinsic backgrounds.

In line with this view, fate divergence of *Cit-k KO* dOPCs and vOPCs appeared to be associated with their developmental origin, i.e., with their derivation from either dorsal Emx1$^+$ or ventral Nkx2.1$^+$ precursors. Additionally, dOPC vs. vOPC alternative fates in oligodendroglia-specific Sox10$^{Cre}$;Cit-k$^{fl/fl}$ mutants strongly support a prominent role of cell-autonomous factors in oligodendroglial alterations. This interpretation also extends to myelin deposition, whose pattern in Sox10$^{Cre}$;Cit-k$^{fl/fl}$ mutants

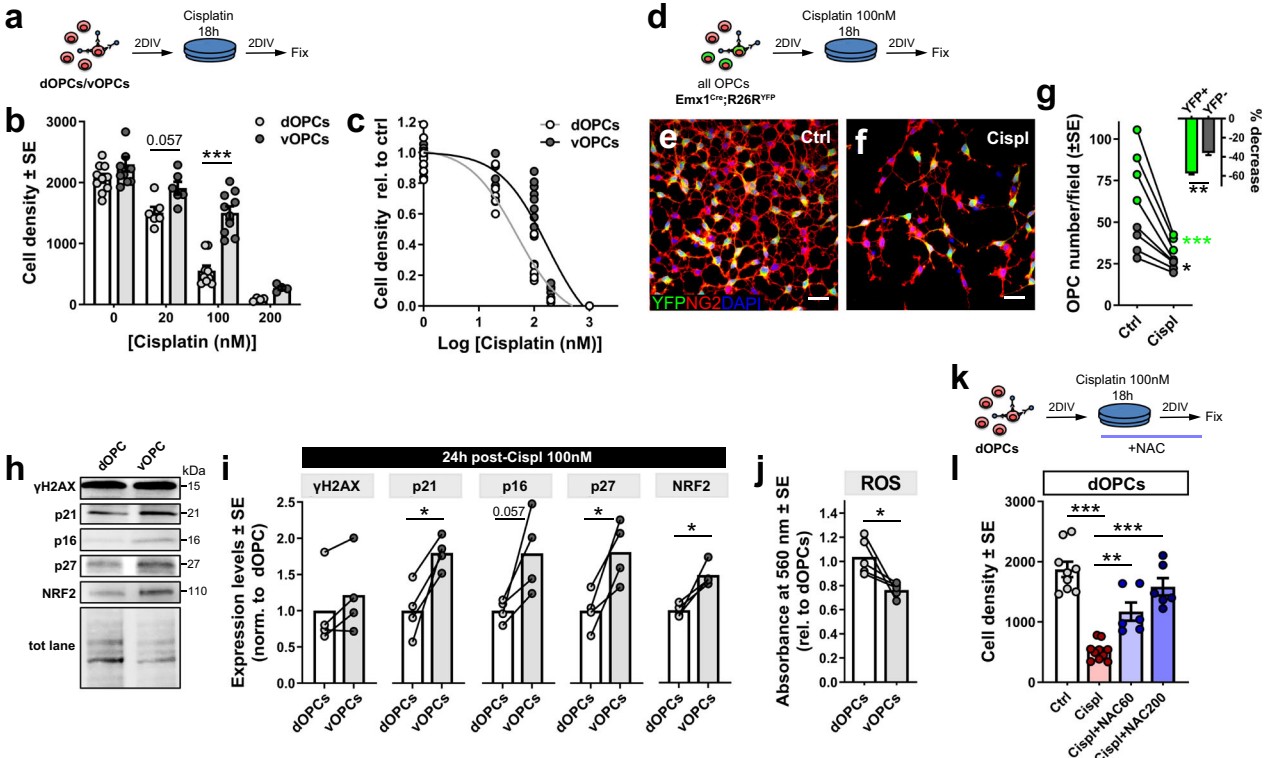

**Fig. 8 WT dorsal OPCs are more vulnerable to DNA damage compared to vOPCs. a** Cisplatin treatment on cultured dOPCs and vOPCs isolated from P8 WT mice. **b** dOPC vs. vOPC density 48 h after cisplatin treatment (0, 20, 100, 200, 1000 nM; Two-way Anova followed by Bonferroni's Multiple Comparison Test, Cispl concentration: $P < 0.0001$, $F(3,51) = 124.2$; OPC origin: $P < 0.0001$, $F(1,51) = 33.53$; Interaction: $P = 0.0006$, $F(3,51) = 6.833$. 4 experiments, dOPC Ctrl = 10 coverslips, vOPC Ctrl = 10, dOPC cispl 20 nM = 6, vOPC cispl 20 nM = 6, dOPC cispl 100 nM = 9, vOPC cispl 100 nM = 10, dOPC cispl 200 nM = 4, vOPC cispl 200 nM = 4, dOPC cispl 1 μM = 4, vOPC cispl 1 μM = 4). **c** Log(inhibitor) vs. response curve (Non-linear regression inhibition curve) representing percentages of viable OPCs 48 h after cisplatin treatment (0, 20, 100, 200, 1000 nM; dOPCs LC50 = 48 nM, $R^2$ = 0.940; dOPCs LC50 = 173.7 nM, $R^2$ = 0.827). **d** Cisplatin treatment on cultured OPCs isolated from P10 Emx1[Cre];R26R[YFP] mouse forebrain. **e, f** YFP+ (green) and YFP-negative NG2+ (red) OPCs in Ctrl (**e**) and Cispl (**f**) condition. DAPI counterstains cell nuclei. Scale bars: 20 μm. **g** Mean YFP+ and YFP-negative OPC number/40× field 48 h after 100 nM cisplatin (Two-way Anova followed by Bonferroni's Multiple Comparison Test, Treatment: $P < 0.0001$, $F(1,6) = 97.68$; Genotype: $P = 0.0049$, $F(1,6) = 18.76$; Interaction: $P = 0.0017$, $F(1,6) = 29.10$. 4 experiments (4 mice) 2 coverslips/mouse per condition). Inset: Mean percentage of YFP+ vs. YFP-negative OPC density decrease (Two-tailed paired t-test, $P = 0.0037$, $t(3) = 8.275$). **h, i** Western blotting analysis of γH2AX, p21, p16, p27, and NRF2 protein expression (relative to total protein lane) in dOPCs vs. vOPCs 24 h after treatment with 100 nM Cispl ($n = 4$/group; Two-tailed Paired t-test: γH2AX: n.s., $t(3) = 2.481$; p21: $P = 0.017$, $t(3) = 4.868$; p16: $P = 0.057$, $t(3) = 2.975$; p27: $P = 0.021$, $t(3) = 4.449$; NRF2: $P = 0.025$, $t(3) = 4.175$). **j** Nitroblue Tetrazolium (NBT) reduction assay in dOPCs vs. vOPCs 24 h after Cispl 100 nM., normalized to dOPC mean value (Two-tailed paired t-test, $P = 0.036$, $t(4) = 3.097$; $n = 5$/group). **k** Cispl + NAC co-treatment experiments on cultured dOPCs isolated from P8 WT mice. **l** NAC supplementation rescues dOPC density upon cisplatin treatment (One-way Anova followed by Bonferroni's Multiple Comparison Test, $P < 0.0001$, $F(3,27) = 31.90$, 5 experiments, dOPC Ctrl = 9 coverslips, dOPC cispl 100 nM = 10, dOPC cispl 100 nM + NAC 60 μg/ml = 6, dOPC cispl 100 nM + NAC 200 μg/ml = 6). In **b, g** inset, **l** data are mean ± SEM. In **g, i, j** lines connect paired samples (i.e., samples from the same mouse). Cispl cisplatin, DIV days in vitro, dorsal oligodendrocyte progenitor cell, N-acetyl cisteine, P postnatal day, vOPC ventral oligodendrocyte progenitor cell, YFP yellow fluorescent protein. *$P < 0.05$; **$P < 0.01$; ***$P < 0.001$. Source data are provided as a Source Data file.

resulted affected also in ventral territories, where OPC density corresponded to that of WT mice. Overall, we propose that cell-autonomous mechanisms associated with OPC developmental origin underlie *Cit-k KO* dOPC vs. vOPC alternative fates in response to the same primary insult (i.e., equivalent accumulation of DNA damage). Of note, populations of mature oligodendrocytes with a different spatial distribution and distinct susceptibility following traumatic spinal cord injury were recently reported. Yet, such vulnerability was not linked to developmental diversity[6], suggesting that local cues or molecular features acquired during lineage progression can override developmental imprinting in mature oligodendrocyte response to injury.

An intrinsic diverse ability to cope with oxidative stress — likely initially part of the DNA damage response[50–52] — appeared to determine the fate of *Cit-k KO* dOPCs vs. vOPCs, as shown by distinct intracellular accumulation of ROS and restoration of

dOPC survival upon NAC treatment. At the molecular level, divergence in the two forebrain *Cit-k KO* OPC subsets emerges downstream to Nrf2 mRNA upregulation. While vOPCs showed a robust anti-oxidant response, low levels of NRF2 protein and of NRF2-target genes were found in dOPCs, This indicates that both *Cit-k KO* OPC subsets can sense increased levels of intracellular ROS, but elicit distinct responses. This can be explained by an altered balance between NRF2 protein production and/or degradation. The relevance of NRF2 dysregulation in *Cit-k KO* dOPC fate was supported by the partial restoration of dOPC in vitro survival obtained by DMF, a positive modulator of NRF2 stability and transcriptional activity[64]. Interestingly, attenuated oxidative stress defenses (i.e., downregulation of SOD and CAT) contributed to cortical OPC apoptosis also in a mouse model of Nijmegen Breakage Syndrome, a genetic disorder characterized by elevated sensitivity to irradiation and microcephaly, caused by the

loss-of-function of NBS, another player in genomic stability[65–67]. Although vOPC phenotype was not investigated in this model, these data further point to the suppression of cytoprotective anti-oxidant responses as a critical event underlying dOPCs susceptibility to DNA damage. Accordingly, in our in vitro assays, WT dOPCs showed reduced expression of NRF2 and a higher ROS-dependent vulnerability to cisplatin-induced DNA damage, compared to vOPCs. Poor ability to cope with oxidative stress upon DNA damage might also explain why dOPCs aged more rapidly[29] and were more heavily affected by irradiation[68], perinatal hypoxia-ischemia[30], and vanadium toxicity[31,32].

The idea that dOPCs — which are normally destined to out-compete vOPCs and persist in most areas of the adult forebrain - are inherently endowed with a lower capacity to counteract oxidative stress might appear paradoxical. However, a prompt ability to repress anti-oxidant systems may be instrumental to activate specific redox-sensitive pathways — e.g., the Wnt and FGF pathways, which are a key player in dOPC specification and amplification[69] — as it was shown for cortical neural progenitors and neuroblasts[70]. Since the development of dOPCs and vOPCs is regulated by distinct extrinsic signals related to dorso-ventral patterning[69], it is conceivable that inherent differences in ROS buffering may be linked to dOPC/vOPC responsivity to different local cues.

Although persisting in the tissue, vOPCs acquired features of senescent cells in Cit-k mutants and after cisplatin treatment. In different types of cancer cells, apoptosis is a response to over-whelming oxidative damage, whereas senescence is a consequence of a less severe insult[44]. Thus, vOPC resilience may be attributed to their better ability to cope with oxidative stress, whose levels are not sufficient to make cells die, but eventually lead them to enter a "dormant" state. Yet, since the senescence mediator p21 can directly bind and stabilize NRF2[71], we cannot exclude that the upregulation of the senescence gene expression program in Cit-k KO vOPC may be instead upstream to the activation of anti-oxidant responses in this OPC subset. Of note, although not showing a lower proliferative ability[21,29], even in absence of any insult, WT vOPCs seem more prone to enter cell senescence compared to later OPC subsets. Higher basal expression of senescence-associated transcripts was found in embryonic fore-brain OPCs compared to their postnatal counterparts[12]. More-over, vOPCs are progressively outnumbered by dOPCs during forebrain development[14,21,25] as well as during the repopulation of OL/OPC-depleted areas[28,29], displaying an inherent dis-advantage that was proposed to be related to a predisposition to undergo replicative senescence[72].

Yet, it remains to be assessed how dOPCs and vOPCs retain the information associated with their lineage. Since the two OPC subsets appear overall transcriptionally equivalent at postnatal stages, this memory might be epigenetically imprinted and result in detectable transcriptional differences only under specific con-ditions, e.g., upon specific stimuli or insults, as also discussed in ref. [11]. MicroRNAs (miRNAs) or long non-coding RNAs (lncRNAs), acting directly on NRF2 (at post-transcriptional levels) or on its interactors[73–75] can be part of a divergent epi-genetic memory[76,77] that dOPCs and vOPCs could inherit from their respective ventricular precursors, as it was shown for neuroblasts[70]. In line with the idea of epigenetic differences in mouse forebrain dOPC vs. vOPCs, the two OPC subsets display a differential propensity to transgress toward other lineages during prenatal development or upon genetic manipulations (i.e., refs. [78–83].). Such differential fate potential has been proposed to arise from the inheritance of a distinct epigenetic memory from early precursors[84].

In conclusion, our data support the idea that, depending on their developmental origin, distinct forebrain OPC subsets are differentially vulnerable to DNA damage and this relies on a divergent modulation of cell anti-oxidant defenses. Elucidating how developmental diversity influences OPC response to injury may be relevant not only to interpret OPC behavior in pathology, but also to provide new ways to approach CNS disorders.

## Methods

**Analyses of human specimens.** Post-mortem immunohistological analyses have been performed on cortical and cerebellar sections of a male newborn (died 1 day after birth) carrying a biallelic truncating variant of CIT-K (CIT-K fs/fs), whose pedigree, genetics, and clinical data are described in ref. [37] (proband B). Luxol Fast Blue was used to highlight myelination defects. Immunostainings were carried out with antibodies directed against γH2AX (S139, 1:100; Cell Signaling Technology, Danvers, MA, USA) and Olig2 (1:500, Millipore, Billerica, MS, USA), according to standard procedures. Parents provided written informed consent for post-mortem studies that followed the approved guidelines of institutional review boards at Great Ormond St. Hospital for Children, London UK, and Children's Hospital of Philadelphia USA.

**Experimental animals.** For histological and molecular analyses, OPC Magnetic-Activated Cell Sorting (MACS) and in vitro functional assays, pharmacological treatments, and intracerebral injection of retroviral particles or Tat-Cre (see below), we employed germinal Cit-k KO mice[33]. Cit-k KO mice survive to approximately postnatal week three, with microcephaly and increased apoptosis of neuronal progenitors largely attributed to the accumulation of DNA damage[34]. Age-matched wild-type (WT) littermates were used as controls. For lineage-tracing analyses in Cit-k KO mice, we crossed Cit-k KO mice with either R26R^YFP (ref. [40] Fig. 3f), Emx1^Cre;R26R^YFP (kindly provided by Prof. Takuji Iwasato, National Institute of Genetics, Japan and Prof. Shigeyoshi Itohara, RIKEN Brain Science Institute, Wako City, Saitama, Japan[85] Fig. 3k) or NG2CreER^TM;R26R^YFP (ref. [41] Fig. 3o) mouse mutants. For transplantation experiments, cells were obtained from β-actin-green fluorescent protein (GFP) mice[86]. Emx1^Cre;R26R^YFP and C57BL/6 J mice were used to study OPC response to cisplatin in vitro. Cit-k^fl/fl mice[45] (ori-ginally obtained from UC Davis KOMP repository as Cit^tm1a(KOMP)Wtsi) were crossed with Sox10^Cre (ref. [87] B6;CBA-Tg(Sox10-cre)1Wdr/J, The Jackson Labora-tory), Emx1^Cre;R26R^YFP or Nkx2.1^Cre;R26R^YFP (ref. [88] C57BL/6J-Tg(Nkx2-1-cre)2Sand/J; The Jackson Laboratory) mouse lines, to conditionally delete Cit-k in oligodendroglia, dorsal progenitors and derivatives, or AEP/POA ventral pro-genitors and derivatives, respectively. These two Cre mouse lines allowed fate map OPCs deriving from non-overlapping dorsal vs. ventral precursor populations. Since in Emx1^Cre and Gsh2^Cre mouse lines targeted cell populations partially overlap (i.e., precursors in the cortico-striatal border are Emx1+/Gsh2+[72]), to avoid possible misinterpretations, Gsh2-derived vOPCs were not investigated in this study.

Experiments and analyses involved mice of both sexes. Groups of 4–5 mice were housed in transparent polycarbonate cages (Tecnoplast, Buggirate, Italy) provided with sawdust bedding, boxes/tunnels hideout as environmental enrichment, and striped paper as nesting material. Food and water were provided ad libitum; environmental conditions were 12 h/12 h light/dark cycle, room temperature 21 ± 1 °C, and room humidity 55 ± 5%.

Perfusions of juvenile and adult mice were carried out under deep general anesthesia obtained by intraperitoneal administration of ketamine (100 mg/kg; Ketavet; Bayern; Leverkusen, Germany) supplemented by xylazine (5 mg/kg; Rompun; Bayer). Postnatal (P0-P10) mice were cryoanesthetized in melting ice. Embryonic day 14 (E14) embryos were obtained from dams under deep general anesthesia. To assess OPC proliferative activity, we employed the thymidine analog 5-bromo-2-deoxyuridine (BrdU, 1 injection i.p. 50 mg/kg body weight; Sigma–Aldrich) that is incorporated in the DNA during the S-phase of the cell cycle.

The experimental plan was designed according to the guidelines of the NIH, the European Communities Council (2010/63/EU), and the Italian Law for Care and Use of Experimental Animals (DL26/2014). It was also approved by the Italian Ministry of Health (authorization 1112/2016-PR to AB and authorization 510/2020-PR to EB) and the Bioethical Committee of the University of Turin. The study was conducted according to the ARRIVE guidelines.

**Histological procedures.** For histological analyses, animals were anaesthetized (as above) and transcardially perfused with 4% paraformaldehyde (PFA) in 0.1 M phosphate buffer (PB). Embryos were maintained in 4% PFA for 12–16 h at 4 °C. Brains were post-fixed for 2 h, cryoprotected, and processed according to standard protocols[89]. Brains were cut in 30 μm-thick coronal sections collected in PBS (for adult/juvenile mice) or on gelatin-coated slides (for embryos) and then stained to detect the expression of different antigens: Olig2 (1:500, Millipore, Billerica, MS, USA); NG2 (1:200, Millipore, Billerica, MS, USA); PDGFRα (APA-5 clone, 1:300, BD Biosciences, San Jose, CA, USA; Eppendorf); CC1 (1:1500, Millipore, Bur-lington, MA, USA); MBP (Smi-99 clone, 1:1000 Sternberger); GFP (1:700, Aves-Labs Inc. Davis CA, USA); cleaved-caspase 3 (1:150, Cell Signaling Technology, Danvers, MA, USA); GFAP (1:1000, Dakopatts, Agilent, Santa Clara, CA); γH2AX

(S139, 1:100; Cell Signaling Technology, Danvers, MA, USA); BrdU (1:500; Abcam, Cambridge, UK); PH3 (1:500, Millipore, Burlington, MA, USA); BLBP (1:200, Millipore, Billerica, MS, USA). GPR17 was detected by means of affinity-purified antibodies (1:100;[38]). To allow BrdU recognition, slices were treated with 2 N HCl for 20 min at 37 °C, followed by 10 min in borate buffer before adding primary antibodies. Incubation with primary antibodies was made overnight at 4 °C in PBS with 0.5% Triton-X 100. The sections were then exposed for 2 h at room temperature (RT) to secondary antibodies: Alexafluor488-conjugated anti-Chicken (1:500, Molecular Probes Life Technologies), anti-Mouse (1:500, Molecular Probes Life Technologies), anti-Rabbit (1:500, Molecular Probes Life Technologies), anti-Rat (1:500, Jackson ImmunoResearch Laboratories, West Grove, PA); Cy3-conjugated anti-Mouse (1:100, Jackson ImmunoResearch Laboratories) and anti-Rat (1:500, Jackson ImmunoResearch Laboratories); Alexafluor555-conjugated anti-Rabbit (1:500, Molecular Probes Life Technologies). 4,6-diamidino-2-phenylindole (DAPI, Fluka, Saint Louis, USA) was used to counterstain cell nuclei. After processing, sections were mounted on microscope slides with Tris-glycerol supplemented with 10% Mowiol (Calbiochem, LaJolla, CA). Myelin silver nitrate Gallyas staining was performed as in ref. [90].

The Senescence-Associated β-Galactosidase (SA-βGAL) Staining Kit (Sigma–Aldrich, Saint Louis, MS, USA) was used to detect β-galactosidase activity at pH 6, a known characteristic of senescent cells, on slices obtained from frozen brain tissue, as described in the manufacturer's instructions. To assess the oligodendroglia identity of SA-βGAL$^+$ cells, SA-βGAL staining was combined with anti-Sox10 (1:1000, Sigma–Aldrich) immunohistochemistry, according to standard procedures. Nissl staining was performed according to standard protocols[34].

**Electron microscopy**. For conventional electron microscopy, P14 *Cit-k KO* and WT mice were perfused transcardially with PB (0.1 M, pH 7.4) followed by 2% PFA and 2% glutaraldehyde in PB, according to ref. [91]. Brains were post-fixed overnight at 4 °C in the same fixative. Vibratome sections (200–400 μm thick) were cut, and post-fixed with 1% osmium tetroxide for 1 h at 4 °C, then stained with uranyl acetate replacement stain (Electron Microscopy Sciences, USA). After dehydration in ethanol, samples were cleared in propylene oxide and embedded in Araldite (Fluka, Saint Louis, USA). Semithin sections (1 μm thick) were obtained at the ultramicrotome (Ultracut UCT, Leica, Wetzlar, Germany), stained with 1% toluidine blue and 2% borate in distilled water, and then observed under a light microscope for precise callosal location. Ultrathin sections (70–100 nm) were examined under a transmission electron microscope (JEOL, JEM-1010, Tokyo, Japan) equipped with a Mega-View-III digital camera and a Soft-Imaging-System (SIS, Münster, Germany) for computerized acquisition of the images.

**Transplantation experiments**. Donor cells for transplantation experiments were isolated from the forebrain of postnatal day 2 (P2) β-actin-GFP donors and grafted into P2 *Cit-k KO* hosts (Supplementary Fig. 4a), according to previously established procedures[92]. Single-cell suspensions were obtained by mechanical dissociation of the nervous tissue and 1 μl of the obtained suspension (final concentration $5 \times 10^4$ cells/μl) was injected through a glass capillary targeting the deep dorsal cortex and striatum of the hosts (2 injections/side). Recipient animals were sacrificed at different survival times post-transplantation, as detailed in Results.

**Lineage-tracing experiments in *Cit-k KO* mice**. To track cells generated from VZ/SVZ dividing progenitors around birth, we employed a retrovirus (RV) derived from the pMIG expression vector carrying a GFP reporter (kind gift of Verdon Taylor, University of Basel). RV can only be incorporated in the genome of mitotic cells, thus permanently labeling their entire progeny. RV particles ($10^9$ TU/ml) were diluted 10:1 ratio of virus:Fast Green (10%) (Sigma–Aldrich, Saint Louis, MS, USA) and bilaterally injected in the lateral ventricles (1 μl/side with borosilicate glass micropapillaries connected to a picopump, WPI) in P0-P1 mice already anesthetized on melting ice. Mice were then sacrificed at P7 (Fig. 3a).

To increase the number of the tagged VZ/SVZ progenitors, a cell-permeant Tat-Cre Recombinase fusion protein (kind gift of Prof. Massimiliano Mazzone, Vesalius Research Center, Leuven, Belgium) was injected as described above in P0-P1 mice obtained by crossing WT and *Cit-k KO* mice with a R26R$^{YFP}$ reporter mouse line[40]. Mice were then sacrificed at P10 (Fig. 3f). To target the entire population of dorsally derived OPCs, we crossed WT/ *Cit-k KO* with Emx1$^{Cre}$;R26R$^{YFP}$ mice (ref. [14] Fig. 3k).

To follow the fate of the progeny of embryonically generated OPCs, we administered 8 mg of tamoxifen to pregnant NG2-CreER$^{TM}$;R26R$^{YFP}$ × WT/ *Cit-k KO* mice at E14 (ref. [41] Fig. 3o) via gastric gavage. Tamoxifen (Sigma) was dissolved in autoclaved corn oil at 20 mg/ml.

**In vivo anti-oxidant treatment**. N-acetyl-L-cysteine (NAC; Sigma–Aldrich, Saint Louis, MS, USA) was reconstituted in PBS to the concentration of 10 mg/ml (pH adjusted to 7.2 with NaOH). 150 mg/kg of NAC was injected in *Cit-k KO* mice daily from P0 to P10 by subcutaneous injections (ref. [93] Fig. 7a, i).

**Magnetic-activated cell sorting (MACS) isolation, cell culture procedures, and in vitro analyses**. After tissue dissociation with the Neural Tissue Dissociation Kit P (Miltenyi Biotech GmbH, Bergisch Gladbach, DE), mouse OPCs were enriched by positive selection using an anti-PDGFRα antibody conjugated to magnetic beads, according to the instructions of the manufacturer (Miltenyi Biotech GmbH, Bergisch Gladbach, DE). Depending on the experiment, MACSorted OPCs were plated onto poly-D-lysine (1 μg/ml, Sigma–Aldrich, Saint Louis, MS, USA) coated glass coverslips in a proliferative medium including Neurobasal, 1× B27 (Invitrogen, Milan, Italy), 2 mM L-glutamine (Sigma–Aldrich, Saint Louis, MS, USA), 10 ng/ml PDGF-BB and 10 ng/ml human bFGF (Miltenyi Biotech GmbH, Bergisch Gladbach, DE), or immediately processed for quantitative RT-PCR analysis or used for functional/biochemical assays. The purity of the MACS-selected OPCs was verified by immunocytochemistry: i.e., more than 95% of the cells were NG2$^+$ at 6 h post-plating and about 98% of the cells showed positivity for PDGFRα and Sox10 24 h post-plating (Supplementary Fig. 10a). As further evidence of the robustness of the MACSorting and of the virtually exclusive oligodendroglia identity of the isolated cells, when we MACSorted cells from Sox10$^{Cre}$;R26R$^{YFP}$ mouse forebrain, the obtained cell population was 100% YFP$^+$ and NG2$^+$ (Supplementary Fig. 10b).

For immunocytochemistry, cells were fixed for 20 min at RT with 4% PFA in 0.1 M PB and labeled with anti-NG2 (1:500, Millipore, Billerica, MS, USA), -γH2AX (S139; 1:500; Cell Signaling Technology, Danvers, MA, USA), -NRF2 (1:200; Abcam, Cambridge, GB), -AN2 (rat homologue of NG2, 1:100; kind gift of Miltenyi Biotech GmbH, Bergisch Gladbach, DE, and Prof. J. Trotter, Johannes Gutemberg University of Mainz, DE), -GFP (1:700, Molecular Probes, Life Technologies, Eugene, Oregon) antibodies overnight at 4 °C in PBS with 0.25% Triton-X. Then, coverslips were incubated with Alexa488- and Alexa555- conjugated secondary antibody (Molecular Probes, Eugene, Oregon) for 1 h RT. After a 5 min incubation with DAPI (1:1000, Fluka, Saint Louis, USA), coverslips were mounted with Tris-glycerol supplemented with 10% Mowiol (Calbiochem, LaJolla, CA).

For determination of oxidative stress, MACSorted OPCs were let adhere for 3 h onto poly-D-lysine coated glass coverslips and then incubated with DHE (Molecular Probes, Eugene, Oregon) at a final concentration of 30 μM in PBS at 37 °C for 5 min. Then, cells were washed with PBS and fixed as described above. Quantitative determination of intracellular ROS by the colorimetric Nitroblue Tetrazolium (NBT) reduction assay was performed as in ref. [54], immediately after WT/*Cit-k KO* OPC MACS-sorting (on equivalent numbers of OPCs/group). Blue NBT formazan deposits were quantified by measuring the absorbance at 560 nm using a microplate reader (Infinite M Nano, Tecan, Männedorf, Switzerland) and converting the obtained raw absorbance values based on a calibration curve built on serial dilutions of WT OPCs.

Expression of NRF2, γH2AX, and Senescence-Associated β-Galactosidase (Senescence-Associated β-Galactosidase Staining Kit; Sigma–Aldrich, Saint Louis, MS, USA) were assessed on MACSorted OPCs after 3 h of adhesion onto poly-D-lysine coated glass coverslips in proliferative medium (see above). To assess cell vulnerability to oxidative stress, immediately after MACS-isolation, OPCs were divided into equivalent aliquots maintained in suspension on slow rotation and incubated in a minimal medium including Neurobasal, B27, and glutamine (Ctrl), supplemented with 100, 500 or 5000 μM hydrogen peroxide (H2O2; Sigma–Aldrich, Saint Louis, MS, USA) for 1 h at 37 °C. Then the medium was replaced with the Ctrl medium and cells were incubated for another 2 h at 37 °C on slow rotation. Trypan blue (Sigma–Aldrich, Saint Louis, MS, USA) was used to determine the number of viable cells.

In a set of experiments, to assess whether counteracting oxidative stress or promoting NRF2 activity could sustain *Cit-k KO* dOPC survival, after MACSorting and plating, dOPCs were cultivated for 48 h in a proliferative medium supplemented with 200 μg/ml NAC[94] or 15 μM dimethyl fumarate[95] (DMF; Sigma–Aldrich, Saint Louis, MS, USA). Ctrl cells were instead treated for 48 h with 0.1% DMSO (vehicle for DMF).

To assess WT OPC vulnerability to chemically-induced DNA damage, OPCs were MACS-isolated from either the dorsal cortex and ventral forebrain of P8 C57BL/6 J WT mice or from the entire forebrain of P10 Emx1$^{Cre}$;R26R$^{YFP}$ mice and plated on coverslips in the proliferative medium at a density adequate to obtain homogenous proliferative rates (50,000 cells/coverslip 12 mm). After 48 h, OPCs were incubated with titrated concentrations (see Fig. 8b) of cisplatin (1 mg/ml stock, Teva Pharmaceuticals, USA) for 18 h. To assess cell survival, cells were then fixed at 48 h after cisplatin removal and immunostained as described above. In a set of experiments, NAC (60 μg/ml[94] or 200 μg/ml) was added to the medium during 100 nM cisplatin treatment and during the following 48 h. To assess protein expression by Western Blot, whole-cell homogenates from OPCs were instead obtained (as described below) 24 h after 100 nM cisplatin/vehicle treatment, when dOPC/vOPC densities did not differ from each other, and compared with the control condition. Similarly, in cisplatin-treated WT d/vOPCs, quantitative determination of intracellular ROS by NBT reduction assay was performed as described above, on equivalent numbers of OPCs/group, at 24 h after 100 nM cisplatin treatment.

For most analyses, OPCs were isolated from P8-P10 mice, in order to obtain a homogeneous dOPC population exclusively derived from Emx1$^+$ progenitors[14].

**Organotypic cultures**. 250 μm thick coronal slices of P10 *Cit-k KO* mouse forebrain and cerebellum were cut using a Tissue Chopper. The slices were placed on Millicell-CM culture plate inserts (Millipore, Billerica, MS, USA) in a medium composed of 50% basal medium with Earle's salts, 25% Hanks' buffered salt solution, 25% horse serum, 5 mg/ml glucose, 0.5 nM triiodothyronine (T3; Sigma–Aldrich, Saint Louis, MS, USA), 60 μg/ml NAC. After 7 days in vitro (DIV),

slices were either fixed for 30 min at RT with 4% PFA in 0.1 M PB for subsequent anti-MBP immunolabelings (see above) or pooled (3 millicells/sample) and lysed for the subsequent protein extraction.

**Quantitative RT-PCR**. Total RNA was extracted with the RNeasy micro kit (Qiagen GmbH, Hilden, DE), and reverse transcribed to cDNA with the High-Capacity cDNA Archive kit (Applied Biosystems, Thermofisher, Waltham, USA). Quantitative Real Time RT-PCR was performed as described in ref. [96], either with predeveloped Taqman assays (Applied Biosystems, Thermofisher, Waltham, USA) or by combining the RealTime Ready Universal Probe Library (UPL, Roche Diagnostics, Monza, Italy) with the primers listed in Supplementary Table 2. Real Time data were collected on the Applied Biosystems StepOnePlus Real-Time PCR System with StepOne™ Software. Data analysis was performed with Microsoft Excel (Microsoft Office 365). A relative quantification approach was used, according to the $2^{-ddCT}$ method[97]. β-actin was used to normalize expression levels.

**Cell lysates and western blotting**. Whole-cell lysates from OPCs were obtained adding 2% SDS for 15 min at 95 °C and from organotypic cultures adding RIPA buffer (1% NP40, 150 mM NaCl, 50 mM TRIS HCl pH 8, 5 mM EDTA, 0.01% SDS, 0.005% Sodium deoxycholate, Roche protease inhibitors, PMSF) for 10 min at 4 °C. Samples were homogenized on ice with a pellet pestle (Sigma–Aldrich, Saint Louis, MS, USA) and centrifuged at $18,000 \times g$ at 4 °C. For immunoblots, equal amounts of proteins from both whole-cell lysates were resolved by SDS–PAGE and blotted to nitrocellulose membranes, which were then probed with rabbit anti-p21 (1:1000, Santa Cruz-MW: 21 kDa), -Phospho-p53 (1:1000, Cell Signaling Technology, Danvers, MS, USA—MW: 53 KDa), -γH2AX (1:1000, Cell Signaling Technology, Danvers, MS, USA—MW: 15 KDa), -NRF2 (1:500, Cell Signaling Technology, Danvers, MS, USA—MW: 120 KDa), -p16 (1:1000, Abcam – MW: 16 kDa) and mouse anti-p53 (1:1000, Cell Signaling Technology, Danvers, MS, USA—MW: 53 kDa), -p27 (1:1000, BD Biosciences, San Jose, CA, USA—MW: 27 kDa), -MBP (1:1000, Millipore, Billerica, MS, USA—MW: 18–21 kDa), -CIT (1:1000, Transduction Laboratories, BD Biosciences, San Jose, CA, USA—MW (Kinase): 225 kDa). The membranes were subsequently incubated with the secondary antibodies and developed using the Clarity substrate (Bio-Rad). Signals are normalized using rabbit anti-glyceraldehyde 3- phosphate dehydrogenase (GAPDH, 1:1000, Cell Signaling Technology, Danvers, MS, USA—MW: 37 kDa), mouse anti-αTubulin (1:5000, Sigma–Aldrich, Saint Louis, MS, USA—MW: 50 kDa), mouse anti-βActin (1:5000, Sigma–Aldrich, Saint Louis, MS, USA—MW: 42 kDa) and Total Lane. Blots were imaged on a ChemiDoc™ (Bio-Rad) and analyzed using Image Lab software 3.0.

**Image processing and data analysis**. Histological specimens were examined using an E-800 Nikon microscope connected to a color CCD Camera, an Axio Scan Z.1 microscope slide scanner (with the associated ZEN software, Zeiss, Oberkochen, Germany), a Nikon C1 confocal microscope (with the associated EZ-C1 Ver3.90 software, Nikon, Melville, NY), and a Leica TCS SP5 confocal microscope (with the associated LAS AF 4.0 software, Leica Microsystems, Wetzlar, Germany). Confocal images (1024 × 1024 pixels) were all acquired at 40×, with a speed of 100–50 Hz and 67.9 μm pinhole size. Adobe Photoshop 6.0 (Adobe Systems, San Jose, CA) was used to assemble the final plates. Quantitative evaluations were performed on confocal images followed by Neurolucida (MicroBrightfield, Colchester, VT) or ImageJ (Research Service Branch, National Institutes of Health, Bethesda, MD; available at http://rsb.info.nih.gov/ij/) analyses. The thickness of the cerebral cortex and ventral forebrain (including striatum and hypothalamus) was measured in coronal sections of WT and *Cit-k KO* mice at P3 and P14 for ANCOVA analysis of the impact of tissue expansion on changes in mutant OPC density. For lineage-tracing analyses exploiting RV and Tat-Cre (see above), the absolute number of targeted (i.e., GFP⁺ or YFP⁺) dSVZ precursors and of dOPC parenchymal derivatives, as well as their ratio, were quantified. Similarly, for lineage-tracing analyses in Emx1^Cre^;R26R^YFP^ × WT/ *Cit-k KO* mice and in NG2-CreER™;R26R^YFP^ × WT/ *Cit-k KO*, the absolute number of targeted (i.e., YFP⁺) NG2⁺ OPCs was quantified. DHE and NRF2 staining intensities were assessed as integrated density (i.e., mean intensity multiplied by the area, including cytoplasm and nucleus) with ImageJ. Since in most *Cit-k KO* OPCs γH2AX immunostaining resulted in almost fully labeled nuclei (Fig. 6a), hampering the identification of single foci, the number of γH2AX⁺ foci was assessed by dividing the stained γH2AX⁺ nuclear area by the mean area sampled on clearly discernible 50 single foci. γH2AX nuclear-integrated density (see above) was also calculated by ImageJ. In all histological quantifications, at least three animals and at least three sections per animal were analyzed for each time point and experimental condition.

**Statistics and reproducibility**. Statistical analyses were carried out with GraphPad Prism 9 (GraphPad software, Inc). The Shapiro-Wilk test was first applied to test for a normal distribution of the data. When normally distributed, unpaired Student's *t*-test (to compare two groups), One-way and Two-ways ANOVA test (for multiple group comparisons) followed by Bonferroni's post-hoc analysis, were used. Alternatively, when data were not normally distributed, Mann–Whitney U-test (to compare two groups), Wilcoxon matched-pairs signed-rank test (to compare to paired groups), and Kruskal-Wallis test followed by Dunn's post-hoc

analysis (for multiple group comparisons) were used. Statistics also included ANCOVA (analysis of covariance), Chi-square test (to compare frequencies), and linear regression analysis (to assess the possible correlation between OPC apoptosis and multinucleation). To assess differences in cell vulnerability to $H_2O_2$ and cisplatin and determine their LC50 and confidence intervals, non-linear regression $\log[H_2O_2/\text{cisplatin}]$/response inhibition curves were built and analyzed with GraphPad Prism 5. In all instances, $P < 0.05$ was considered statistically significant. Histograms represent mean ± standard error (SE). Statistical differences were indicated with *$P < 0.05$, **$P < 0.01$, ***$P < 0.001$. The list of the applied tests in each case, F and n (animals, samples) values, and the number of experiments are included in Supplementary Table 1. Each experiment was repeated at least three times independently with similar results. For tissue analyses, mice were obtained from more than two different litters. Representative immuno-/histochemical staining images are representative of at least three sections from at least three different animals/groups. Uncropped blots are provided in the Source Data file.

**Reporting summary**. Further information on research design is available in the Nature Research Reporting Summary linked to this article.

## Data availability
Data supporting the findings in this paper are available from the corresponding author upon request. Source data are provided with this paper.

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

## Acknowledgements

We wish to thank Prof. Verdon Taylor (University of Basel) for the kind gift of RV-GFP; Prof. Massimiliano Mazzone (Vesalius Research Center, Leuven, Belgium) for the kind gift of the Tat-Cre; Prof. Takuji Iwasato (National Institute of Genetics, Japan) and Prof. Shigeyoshi Itohara (RIKEN Brain Science Institute, Wako City, Saitama, Japan) who kindly provided the Emx1Cre;R26RYFP mouse line; Miltenyi Biotech GmbH and Prof. Jacqueline Trotter (Johannes Gutemberg University of Mainz, Germany) for sharing anti-AN2 antibody. We wish to thank Dr. Cecilia Astigiano and Dr. Maryam Ardakani Khastkhodaei for technical support. Our work has been supported by: the "Cariplo Ricerca Biomedica Giovani Ricercatori" grant from the Cariplo Foundation (ID: 2014–1207), the Cassa di Risparmio di Torino (CRT) Foundation grant (ID 2021.0657), Individual funding for basic research (Ffabr) granted by the Italian Agency for the Evaluation of University and Research, and local funds of University of Turin, to E.B.; Merck Serono Grant GMSI 2015, funds of the University of Turin and Compagnia di San Paolo (S1618 Grant), individual funding for basic research (Ffabr) granted by the Italian Agency for the Evaluation of University and Research, and local funds of the University of Turin, to A.B. This study was also supported by Ministero dell'Istruzione, dell'Università e della Ricerca—MIUR project "Dipartimenti di Eccellenza 2018–2022" to Dept. of Neuroscience "Rita Levi Montalcini".

## Author contributions

E.B. and A.B.: Conceptualization, data curation, formal analysis, investigation, methodology, visualization, writing—original draft, writing—review and editing, funding acquisition, project administration, supervision. M.L., R.P., B.H., and G.P.: Formal analysis, investigation, methodology, visualization, writing—review and editing. L.B., A.M., S.B., and F.D.C.: Methodology, writing—review and editing.

## Competing interests

The authors declare no competing interests.
