## [Peer Review File · Nature Communications]

Molecular and functional heterogeneity in dorsal and ventral oligodendrocyte progenitor cells of the mouse forebrain in response to DNA damageReviewers' comments:

Reviewer #1 (Remarks to the Author):

The question of whether there is true functional heterogeneity in the oligodendrocyte lineage is, as the authors indicate, a topic of much current interest in the (rapidly expanding) oligodendrocyte field. Here the authors provide some interesting and for the most part convincing data that, in the absence of citron-kinase (Cit-K), cells of dorsal or ventral development origin respond differently to the ensuing accumulation of DNA damage - undergoing either apoptosis (dorsal origin) versus senescence (ventral origin). The wider biological relevance of this observation is not clear and the extent to which this has a bearing on the oligodendrocyte lineage in the absence of a disturbance of Cit-K (that is, in normal physiology) is hard to assess. Nevertheless, the results do provide further confirmatory evidence that there are intrinsic differences in function that relate to developmental origin and are therefore indicative of functional heterogeneity within the lineage.

Here are some specific and minor points that should be addressed.

OPC = oligodendrocyte progenitor cell - not precursor cell.

Abstract - "Whether developmental heterogeneity influences OPC behavior and molecular features in pathology is largely unknown." Really? See Crawford et al., Cell Reports 2016 (cited ref 21). I suggest the authors re-write this sentence to acknowledge that functional heterogeneity in the regenerative response to injury has been clearly demonstrated. It does not enhance the interest of the current study to overlook this - rather, it diminishes it.

Abstract - "Here we show that in vivo ablation of Citron-kinase, leading to accumulation of DNA damage, disrupts OPC fate resulting in cell death and senescence in the dorsal and ventral subsets, respectively. This divergence correlates with differential activity of NRF2-mediated anti-oxidant response to DNA lesions in dorsal and ventral OPCs." As written, it is not clear what the divergence is.

Lines 46/47 - replace precursor with progenitor

Line 46 - NG2 is a rather outdated marker for OPCs - many in the field no longer refer to them as NG2 cells, not least because this is now recognised as not being an OPC specific marker.

Line 64 - what is meant by 'scattered data'?

Line 72 - dys-myelination rather than dis-myelination

Line 97 - delete 'consistently'

Lines 98-102 - the text here does not really relate to this study's results and should appear elsewhere - discussion perhaps.

Line 103 - "Phenocopy of myelin and oligodendrocyte alterations were detected in Cit-k KO brain and CIT-K fs/fs post-mortem cortical tissue." - not at all clear what is meant by this sentence. Please re-write more clearly

Line 110 - I think the authors means hypo-myelination rather than dys-myelination

Line 116 - the authors need to use an additional marker to NG2 (perhaps olig2 to confirm oligodendrocyte lineage - or PDGFRA as used elsewhere)

Line 117 - not clear whether it is mouse or human material that is being analysed

Line 132 - Please re-write this sentence - not clear what point is being made.

Line 175 - perhaps change 'scenario' to 'interpretation'?

Line 273 - Re-phrase this section sub-heading

Generally, the text would benefit from some re-writing (in addition to specific suggestions made). For example, remove words like 'importantly' and "notably" - which are responses that should be left for the reader.

The increased susceptibility of dOPCs to DNA damage might also explain why they age more rapidly (Crawford et al., 2016). Perhaps this could be mentioned in the discussion

Reviewer #2 (Remarks to the Author):

In this study, Boda et al. attempted to address molecular and functional heterogeneity of OPCs in dorsal or ventral regions in brain by using Citron-kinase KO mice, which develop microcephaly. They found that the extent of the OPC loss was different between dorsal and ventral forebrain in Cit-K KO mice, and that the dorsal and ventral OPCs exhibited differential response to DNA damages in vitro. They would like to draw a conclusion that the difference of dorsal and ventral OPCs is due to their differential responses to oxidative stress. Treatment with an anti-oxidant NAC could partially rescue the oligodendroglia phenotype. Overall, the study is quite descriptive without defining mechanisms underlying the difference between dorsal and ventral OPCs and their differential responses to DNA damage responses and oxidative stress. The global KO of Cit-K impairs neurogenesis, gliogenesis and neuronal functions, it is not clear to what extent the defects or changes in brain development and microenvironment affect dOPC and vOPC behaviours in the Cit-k KO mice. There is also a lack of in vivo evidence showing that dorsal OPCs undergo cell death while ventral OPCs undergo senescence in the Cit-K KO brain. In addition, many experimental data are not convincing and even conflicting and should be further validated.

Critiques:

1. The major point of the study is to address molecular and functional heterogeneity of OPCs. However, this study is on the basis of global Cit-k KO mice, and there is no data to validate their conclusion of dorsal and ventral OPC heterogeneity using dorsal and ventral specific Cit-k KO mice. The global KO of Cit-K impairs neurogenesis, gliogenesis and neuronal functions, it is unclear to what extent the defects in brain progenitor cell differentiation and microenvironment changes in the global Cit-k KO contribute to behaviour differences between dOPCs and vOPCs during development. Without region-specific KO analysis, one could hardly draw the conclusion of dorsal and ventral OPC heterogeneity.
2. The authors claimed that there is more cell apoptosis in dorsal OPCs than ventral OPCs. However, they did not present in vivo data in the brain e.g. by immunohistochemistry.
3. The senescent phenotype of vOPCs in Fig 4 was not confirmed in the brain of Cit-k KO mice.
4. In Fig 5a-d, the authors showed that Cit-k KO dOPCs and vOPCs display the same expression levels of DNA damage response marker γ H2AX, however, DNA damage response-induced ROS between dOPCs and vOPCs is different (Fig. 5c,d). The measurement of DHE is based on immunofluorescence intensity, which is not reliable for quantification.
5. There are no experimental studies to address the mechanisms for differential ROS responses between dOPCs and vOPCs in Cit-k KO mice.
6. Cit-kinase KO results in DNA damage responses and oxidative stress in OPCs. It is unclear whether these events are directly regulated by Cit kinase or secondary to the Cit-kinase loss-induced cell death responses.
7. qPCR measurement of Nrf2 expression did not show the difference between dorsal OPCs and ventral OPCs in Fig. 5h, however, in figure 5k, the authors used the immunostaining intensity to quantify the NRF2 expression level. This kind of quantification is not convincing. They should do

western blot analysis for the sorted cells.

8. In the Figs 6 and 7, it is intriguing that NAC treatment rescued the OPC loss phenotype of Cit-k KO mice. The authors used organotypic culture but did not show if there is any increase of mature oligodendrocytes in the treated brain. In addition, they did not show if NAC treatment could rescue DNA damage responses in OPCs caused by Cit-k KO.

9. The authors did not show whether treatment with NAC inhibits ROS responses e.g. DHE in Cit-k KO OPCs? Does NAC treatment increase OPC survival and proliferation from Cit-k KO?

10. They isolated cells from P8 Emx1Cre-R26RYFP mouse dorsal cortex and ventral forebrain. Emx1Cre is mainly restricted to dorsal neural progenitors. It is not clear why there are many Emx1CreGFP+ cells in vOPCs (Fig 7d). MACS sorting with PDGFR α antibody could also isolate other cell types like NG2+ pericytes. The authors did not validate that the isolated NG2+ cells are OPCs.

11. The authors indicate that p53 is correlated with NRF2 stability. The authors showed p21 upregulation in ventral OPCs, they should show p53 and phospho-p53 expression in Cit-k KO brain. Does NAC treatment alter p53 and p21 expression in the Cit-k KO brain?

12. In the Suppl. Fig 2 the authors used NG2CreERTM;R26RYFP to conclude that embryonically generated OPCs were not affected by Cit-k KO. However, in the text (Page 7 line 158-163, also fig 3q), they stated that embryonically generated OPCs decreased in Cit-k KO. Authors should clarify the discrepancy.

Reviewer #3 (Remarks to the Author):

This paper by Boda and colleagues entitled "Citron-kinase deletion unveils inherent 1 molecular and functional heterogeneity in dorsal and ventral oligodendrocyte precursor cells of the mouse forebrain" investigates the effect of genetic deletion of Citron-kinase (Cit-k), a mouse model of human microlissencephaly associated with perinatal lethality in humans and mice. Cit-k has previously been demonstrated to be required for mitosis, organization of the midbody in late cytokinesis and in DNA damage control (doi:10.1242/jcs.200253). Boda and colleagues reveal that Cit-k KO mice exhibit a dramatic loss of pre-myelinating OLs, oligodendrocytes and myelin during postnatal development. They provide evidence of a generalized reduction in the density of OPCs in the Cit-k KO forebrain. The reduction in OPC density was greater for dorsal OPCs than ventral OPCs, with the former exhibiting a progressive reduction from P3 to P14, unlike ventral OPCs which maintained the same density at both time-points. The regional differences in OPC response to Cit-k deletion are ascribed to cell autonomous differences in the response of dorsal and ventral OPCs to DNA-damage induced oxidative stress that are dependent on the developmental origin of the OPCs. They provide evidence that dorsal OPCs that derive from Emx1-progenitors exhibit greater vulnerability to Cit-k deletion and undergo apoptotic cell death. By contrast, ventrally-derived OPCs were more resilient to DNA damage, undergoing senescence rather than apoptosis, which was associated with the upregulation of Nuclear factor erythroid 2-related factor 2 (NRF2), a key mediator of antioxidant defense. They also demonstrate that antioxidant supplementation to Cit-K KO mice confer resistance of dorsal OPCs to DNA damage.

The interest in this work lies in the molecular and functional differences in OPCs of differing developmental origin. The study builds on prior work demonstrating heterogeneity in the function of OPCs that derive from different progenitor cell populations, here demonstrating that dorsal and ventral OPCs differ in their response to DNA damage. The intrinsic differences in dorsal and ventral OPCs to DNA damage has implications for understanding the susceptibility of dorsal OPCs to perinatal hypoxia-ischemia and ageing.

Overall, the study is generally well conducted and the results are important, but this reviewer has a number of concerns that the authors should address to provide more compelling evidence for their conclusions.

Specific concerns:

The authors conclude that “dorsal oligodendrogenesis is not affected by loss of Cit-k” (lines 149-150) and that “normal numbers of OPCs are generated postnatally in the dorsal cortex but undergo a progressive depletion” (lines 164-165). This is based on density measures of fate-mapped OPCs. To reach this conclusion the authors should clarify the labeling efficiency of stem/progenitor cells in the lateral ventricles of WT and Cit-k KO mice using their genetic labeling approaches. Moreover, presenting the results as a density measure alone is misleading since the Cit-k KO brain is significantly smaller than WT. The authors should also provide the total number of neural stem/progenitor cells genetically tagged and the total number cell of YFP+/PDGFRa+ OPCs across the corpus callosum and cerebral cortex. The combination of a genetic approach with EdU pulse/chase labeling would facilitate this analysis.

Line 145-146: Please clarify the meaning of the statement “90% of YFP+/PDGFRa+ OPCs residing in the dorsal cortex/CC of both WT and Cit-k KO mice”. The figure illustrates extensive labelling of cells along the walls of the lateral ventricle but little labeling in the CC or cortex.

The analysis of ventral OPC density is insufficient to reach the conclusion that vOPC genesis and distribution is largely unaffected in the Cit-k KO. To support their conclusion, the authors refer to Supplementary Figure 2 which shows that there is no significant difference in the percentage of YFP+NG2+ OPCs in dorsal, lateral and ventral forebrain in NG2CreERTM;R26RYFP WT and Cip-k KO mice. The genetic labelling approach is not adequately analysed to form a clear conclusion. Most importantly, they do not support the conclusion that there are differential effects on OPCs in dorsal versus ventral forebrain. The analysis also does not take into account differences in OPC density at the time of tamoxifen treatment and the number or density of OPCs at the time of analysis (P10).

To authors state that “dOPCs underwent apoptosis with higher frequencies compared to vOPCs, despite uniform cytokinesis defects across both populations”. Although the authors provide photomicrographs illustrating incomplete binucleated OPCs in both dorsal and ventral forebrain, the abundance relative to mononucleated OPCs is not quantified. Multinucleated OPCs of the type characterized would also be expected of normal cytokinesis in WT mice. Quantification of the percentage of OPCs that are binucleated in WT versus Cip-k KO in both dorsal and ventral forebrain would be required to support any conclusion about cytokinesis defects.

Lines 220-222: The authors state that “the acquisition of senescent properties in Cit-k KO vOPCs depended on cell-intrinsic factors, rather than on environmental senescence-inducers” is not supported by any analysis of senescent markers in the Cit-k KO recipient of GFP+ wildtype telencephalic cells. This would require an examination of a senescence marker such as SA-Bgal in GFP+ cells in the ventral forebrain.

Figure 5. The authors should provide the micrographs of WT P10 OPCs stained with gamma-H2AX.

The assessment of apoptosis by activated caspase-3 labeling was performed only on OPCs isolated from dorsal cortex and corpus callosum of Cit-KO mice. To argue that dOPCs are more vulnerable to apoptosis a comparative analysis of apoptosis in dOPCs and vOPCs should be conducted.

The referring figure number needs to be added to Supplementary Table 1 row 1.

We thank the Reviewers for the supportive statements and the raised criticisms that we believe we have now fully addressed.

Briefly, in response to Reviewer #2's comments, we generated and analyzed three novel conditional *Cit-k* mutant mouse lines to provide evidence of the cell autonomous differential susceptibility of dorsal (dOPCs) and ventral OPCs (vOPCs) to DNA damage, and to link OPC fates with their diverse developmental origin (i.e. from *Emx1+* vs. *Nkx2.1+* precursors). We also extended the investigation of the molecular alterations associated with d/vOPC differential vulnerability by including additional experimental approaches, and assessed the functional relevance of oxidative stress and NRF2 suppression with a pharmacological approach. Moreover, prompted by Reviewer #1's comments, we have strengthened the evidence that, in a condition of DNA damage unrelated to *Cit-k* loss (i.e. upon treatment with a cross-linking agent such as cisplatin), depending on their developmental origin, WT dOPCs vs. vOPCs activate different responses whose molecular bases are shared with *Cit-k KO* d/vOPCs. Finally, we tackled Reviewer#3's criticisms about lineage tracing analyses, multinucleation and cell senescence in grafted cells.

These additional experiments/analyses fully corroborated our conclusions that i) cell-autonomous factors associated with dorsal vs ventral developmental origin underlie dOPC vs vOPC fate divergence (i.e. cell death in dOPCs, cell senescence in vOPCs); ii) the different vulnerability of dOPCs vs. vOPCs to DNA damage relies on a different ability to set up cell antioxidant defenses.

Changes in the text have been highlighted with red characters.

Please, find below the point-by-point answers to the Reviewers' comments.

Reviewers' comments:

Reviewer #1 (Remarks to the Author):

The question of whether there is true functional heterogeneity in the oligodendrocyte lineage is, as the authors indicate, a topic of much current interest in the (rapidly expanding) oligodendrocyte field. Here the authors provide some interesting and for the most part convincing data that, in the absence of citron-kinase (Cit-K), cells of dorsal or ventral development origin respond differently to the ensuing accumulation of DNA damage - undergoing either apoptosis (dorsal origin) versus senescence (ventral origin). The wider biological relevance of this observation is not clear and the extent to which this has a bearing on the oligodendrocyte lineage in the absence of a disturbance of Cit-K (that is, in normal physiology) is hard to assess. Nevertheless, the results do provide further confirmatory evidence that there are intrinsic differences in function that relate to developmental origin and are therefore indicative of functional heterogeneity within the lineage.

We thank the Reviewer for this comment that prompted us to perform novel experiments aimed at increasing the relevance of our findings in *Cit-k* KO models and at showing that:

1. the distinct dorso-ventral vulnerability to DNA damage is a cell-intrinsic OPC feature associated with OPC developmental origin, detectable also in conditions of altered physiology unrelated to *Cit-k* loss such as treatment with the cross-linking agent cisplatin in vitro;
2. molecular alterations associated with dOPCs vs. vOPCs diverse responses to *Cit-k* deletion are shared by WT dOPCs vs. vOPCs responding to DNA damage.

Specifically, in this revised version of the manuscript, we include data showing that:

- when OPCs are isolated from the entire *Emx1*^{Cre};R26R^{YFP} mouse forebrain and YFP⁺ (i.e. dorsally-derived) and YFP-negative (i.e. ventrally-derived) are cultured together, YFP⁺ OPCs show a higher decrease in cell number after cisplatin treatment, compared to YFP-negative OPCs (Fig. 8d-g);
- 24h after cisplatin treatment (i.e. before exhibiting differences in cell survival), WT dOPCs show lower expression of NRF2 and higher intracellular ROS compared to WT vOPCs, similar to what observed in *Cit-k* KO dOPCs vs. vOPCs (Fig. 8h-j);
- 24h after cisplatin treatment, WT vOPCs display higher levels of senescence markers, such as p21, p16 and p27, compared to WT dOPCs, suggesting they are activating a senescence program similar to *Cit-k* KO vOPCs (Fig. 8h-i);
- Similar to what observed in *Cit-k* KO dOPCs, NAC treatment results in an almost complete rescue of WT dOPC survival after cisplatin treatment, suggesting that oxidative stress plays a major role in (WT and KO) dOPC vulnerability to DNA damage (Fig. 8k,l).

Here are some specific and minor points that should be addressed.

1) OPC = oligodendrocyte progenitor cell – not precursor cell.

We have amended the text following the Reviewer's suggestion.

2) Abstract - "Whether developmental heterogeneity influences OPC behavior and molecular features in pathology is largely unknown." Really? See Crawford et al., Cell Reports 2016 (cited ref 21). I suggest the authors re-write this sentence to acknowledge that functional heterogeneity in the regenerative response to injury has been clearly demonstrated. It does not enhance the interest of the current study to overlook this - rather, it diminishes it.

We thank the Reviewer for this remark. Following the Reviewer's suggestion, we have now rephrased this sentence, as follows: "In the developing mouse forebrain, temporally distinct waves of oligodendrocyte progenitor cells (OPCs) arise from different germinal zones and eventually populate either dorsal or ventral regions, where they present as transcriptionally and functionally equivalent cells. Despite that, developmental heterogeneity influences adult OPC response upon demyelination. Whether this occurs also in other injury conditions and the molecular mechanisms that underlie OPC functional differences remain to be investigated." (lines 21-26).

3) Abstract - "Here we show that in vivo ablation of Citron-kinase, leading to accumulation of DNA damage, disrupts OPC fate resulting in cell death and senescence in the dorsal and ventral subsets, respectively. This divergence correlates

with differential activity of NRF2-mediated anti-oxidant response to DNA lesions in dorsal and ventral OPCs.” As written, it is not clear what the divergence is.

Following the Reviewer’s criticism and based on new data included in the revised manuscript, we have now rephrased as follows: “Such alternative fates are associated with OPC diverse developmental origins and with a different activation of NRF2-mediated anti-oxidant responses.” (lines 29-31).

4) Lines 46/47 - replace precursor with progenitor

We have amended the text following the Reviewer’s suggestion: “The diversity of oligodendrocyte progenitor cells (OPCs) is instead under debate.” (lines 41-42).

5) Line 46 - NG2 is a rather outdated marker for OPCs - many in the field no longer refer to them as NG2 cells, not least because this is now recognised as not being an OPC-specific marker.

We agree with the Reviewer that NG2 can be a promiscuous marker. In the current version of the manuscript, we do not mention NG2 anymore as an OPC-specific marker in the introduction. Moreover, we have included quantifications obtained with PDGFR α (lines 125-127 in the Results).

6) Line 64 - what is meant by ‘scattered data’?

We apologize for the unclear formulation. We meant “data found in papers focusing on different topics/injury conditions”. For the sake of clarity, we now have written just “data”. Lines 60-62: “Thus, OPCs with distinct origins appear functionally equivalent. However, data suggest that developmental heterogeneity may influence OPC behavior during aging and their susceptibility/response to injury”.

7) Line 72 - dys-myelination rather than dis-myelination

We have amended the text following the Reviewer’s suggestion. Lines 68-70: “Thus, OPC developmental diversity may contribute to different regional manifestations of de-/dys-myelinating diseases”.

8) Line 97 - delete ‘consistently’

We have amended the text following the Reviewer’s suggestion. Lines 95-97: “Consistent with *CIT-K* participation in DNA damage repair³⁴, widespread phospho-histone H2AX (γ H2AX) labeling, a marker of DNA breaks³⁵, was detected throughout *Cit-k* KO mouse forebrain at postnatal day (P) 14 (Fig.1a-b)”.

9) Lines 98-102 - the text here does not really relate to this study’s results and should appear elsewhere - discussion perhaps.

Following the Reviewer’s suggestion, we have now moved that sentence in the Methods section (as a justification/description of the model). Lines 792-795: “..., we employed germinal *Cit-k* KO mice³³. *Cit-k* KO mice survive to approximately postnatal week three, with microcephaly and increased apoptosis of neuronal progenitors largely attributed to accumulation of DNA damage³⁴.”

10) Line 103 - “Phenocopy of myelin and oligodendrocyte alterations were detected in *Cit-k* KO brain and *CIT-K* fs/fs post-mortem cortical tissue.” - not at all clear what is meant by this sentence. Please re-write more clearly

Following the Reviewer’s request, we have now rephrased as follows: “Myelin and oligodendrocyte alterations were detected in both *Cit-k* KO brain and *CIT-K* fs/fs post-mortem cortical tissue.”(lines 101-102).

11) Line 110 - I think the authors means hypo-myelination rather than dys-myelination

Yes, we agree with the Reviewer and have amended the text accordingly. Lines 108-109: “Likewise, severe hypomyelination of subcortical and cerebellar white matter was observed in *CIT-K* fs/fs tissue (Fig. 1g-j).”

12) Line 116 - the authors need to use an additional marker to NG2 (perhaps olig2 to confirm oligodendrocyte lineage - or PDGFRA as used elsewhere).

We agree with the Reviewer that these results needed to be corroborated by additional analyses based on another OPC-specific marker. Thus, we quantified the abundance of PDGFR α + cells and obtained results consistent with those reported for NG2+ cells. These data are included in the text (lines 125-127: “Similar results were also obtained by counting cells labeled by other OPC markers, such as PDGFR α (412.33 \pm 36.11 WT vs. 56.88 KO cells/mm² in the dorsal cortex; 480.57 \pm 17.89 WT vs. 312.34 \pm 27.93 KO cells/mm² in the POA).

13) Line 117 - not clear whether it is mouse or human material that is being analysed

We thank the reviewer for pointing this out. For the sake of clarity, now the sentence is: “OPCs were evaluated to investigate the origin of the myelination deficit in *Cit-k KO* mouse brain” (lines 113-114).

14) Line 132 - Please re-write this sentence - not clear what point is being made.

We apologise for the unclear formulation of the sentence. Following the Reviewer’s request, we have now rephrased as follows: “The observed dorso-ventral gradient of OPC density was not explained by a different postnatal expansion of *Cit-K KO* dorsal vs ventral regions containing a stable OPC number (ANCOVA analysis, Suppl. Table 1)” (lines 128-130).

15) Line 175 - perhaps change ‘scenario’ to ‘interpretation’?

Following the Reviewer’s suggestion, we have now changed to “interpretation”. Lines 176-180: “The preferential decline of *Cit-k KO* dOPCs vs vOPCs may reflect a differential expression of CIT-K in the two cell subsets. However, equal levels of *Cit-k* mRNA and CIT-K protein detected in WT forebrain dOPCs (i.e. isolated from the dorsal cortex and CC) and vOPCs (i.e. isolated from subcortical regions, including septum, striatum, thalamus and hypothalamus; Suppl. Fig. 3a,b) argued against this interpretation”.

16) Line 273 - Re-phrase this section sub-heading

For the sake of clarity, the subheading was changed to “WT dorsal and ventral OPCs set up different responses to DNA damage *in vitro*” (lines 277-278).

17) Generally, the text would benefit from some re-writing (in addition to specific suggestions made). For example, remove words like ‘importantly’ and “notably” - which are responses that should be left for the reader.

We thank the Reviewer for this remark. We have amended the text following the Reviewer’s suggestion and removed “notably, importantly, of note” adverbs.

18) The increased susceptibility of dOPCs to DNA damage might also explain why they age more rapidly (Crawford et al., 2016). Perhaps this could be mentioned in the discussion

We appreciate the Reviewer’s advice. This comment is now included in the discussion (lines 438-440): “Poor ability to cope with oxidative stress upon DNA damage might also explain why dOPCs aged more rapidly²⁹ and were more heavily affected by irradiation⁶⁸, perinatal hypoxia-ischemia³⁰ and vanadium toxicity^{31,32}.”

Reviewer #2 (Remarks to the Author):

In this study, Boda et al. attempted to address molecular and functional heterogeneity of OPCs in dorsal or ventral regions in brain by using Citron-kinase KO mice, which develop microcephaly. They found that the extent of the OPC loss was different between dorsal and ventral forebrain in Cit-K KO mice, and that the dorsal and ventral OPCs exhibited differential response to DNA damages in vitro. They would like to draw a conclusion that the difference of dorsal and ventral OPCs is due to their differential responses to oxidative stress. Treatment with an anti-oxidant NAC could partially rescue the oligodendroglia phenotype. Overall, the study is quite descriptive without defining mechanisms underlying the difference between dorsal and ventral OPCs and their differential responses to DNA damage responses and oxidative stress. The global KO of Cit-K impairs neurogenesis, gliogenesis and neuronal functions, it is not clear to what extent the defects or changes in brain development and microenvironment affect dOPC and vOPC behaviours in the Cit-k KO mice. There is also a lack of in vivo evidence showing that dorsal OPCs undergo cell death while ventral OPCs undergo senescence in the Cit-K KO brain. In addition, many experimental data are not convincing and even conflicting and should be further validated.

Critiques:

1. The major point of the study is to address molecular and functional heterogeneity of OPCs. However, this study is on the basis of global Cit-k KO mice, and there is no data to validate their conclusion of dorsal and ventral OPC heterogeneity using dorsal and ventral specific Cit-k KO mice. The global KO of Cit-K impairs neurogenesis, gliogenesis and neuronal functions, it is unclear to what extent the defects in brain progenitor cell differentiation and microenvironment changes in the global Cit-k KO contribute to behaviour differences between dOPCs and vOPCs during development. Without region-specific KO analysis, one could hardly draw the conclusion of dorsal and ventral OPC heterogeneity.

We thank the reviewer for this comment and agree that in vivo analyses in the global *Cit-K KO* mouse and ex-vivo experiments on cells derived from this model do not suffice to conclude that *Cit-k KO* dorsal (dOPCs) and ventral OPCs (vOPCs) are intrinsically different in their response to DNA damage and that this is related to their developmental origin.

1a) To assess whether the observed alternative response of *Cit-k KO* dOPCs vs. vOPCs was due to cell intrinsic factors, we generated a conditional Sox10^{Cre};Cit-k^{fl/fl} mouse line to delete *Cit-k* exclusively in oligodendroglia. At P14, a dorso-ventral gradient of OPC abundance - resembling the phenotype of the germinal *Cit-k KO* mouse brain - was present. Similar to *Cit-k KO* dOPCs, Sox10^{Cre};Cit-k^{fl/fl} dOPCs showed higher apoptotic fractions, whereas Sox10^{Cre};Cit-k^{fl/fl} vOPCs showed expression of senescence markers (i.e. *p21* and *Ecrg4* mRNAs, and SA-βGAL-positivity in vivo) (Fig. 5a-e). Expression of the myelin-associated protein MBP was also perturbed in P14 Sox10^{Cre};Cit-k^{fl/fl} forebrain, with a clear overall reduction, especially evident in dorsal areas (Suppl. Fig. 5a,b). These data indicate a cell-intrinsic role of *Cit-k* loss in producing divergent dOPCs vs vOPC alterations.

1b) To assess whether the alternative response of *Cit-k KO* dOPCs vs. vOPCs was linked to their developmental origin, we generated Emx1^{Cre};R26R^{YFP};Cit-k^{fl/fl} and Nkx2.1^{Cre};R26R^{YFP};Cit-k^{fl/fl} mouse lines to delete *Cit-k* in two non-overlapping dorsally-derived or ventrally-derived (1st wave) cell populations, respectively. Since studies showed a partial overlap between targeted cell populations in Emx1^{Cre} and Gsh2^{Cre} mouse lines (i.e. precursors in the cortico-striatal border are Emx1⁺/Gsh2⁺; Naruse et al., 2016; Boshans et al., 2020), to avoid possible misinterpretations, Gsh2-derived vOPCs have not been investigated in this study. At P14-P16, the density of dorsally-derived YFP⁺ OPCs was significantly reduced and higher fractions of apoptotic OPCs were detected in Emx1^{Cre};R26R^{YFP};Cit-k^{fl/fl} mouse dorsal cortex and CC, compared to Emx1^{Cre};R26R^{YFP} controls. In contrast, the density of ventrally-derived YFP⁺ OPCs did not differ in P14-P16 Nkx2.1^{Cre};R26R^{YFP};Cit-k^{fl/fl} vs Nkx2.1^{Cre};R26R^{YFP} control POA (the only region in which YFP⁺ OPCs could be detected at this stage). These data strengthen the link between d/vOPC fates and their lineages. Of note, the overall OPC density did not vary in Emx1^{Cre};R26R^{YFP};Cit-k^{fl/fl} compared to controls, indicating a compensatory expansion of ventrally derived WT OPCs at dorsal sites and further arguing against a major contribution of environmental factors in *Cit-k KO* dOPC depletion/defects (Fig. 5f-j) In line with this interpretation, despite reduced in size, Emx1^{Cre};R26R^{YFP};Cit-k^{fl/fl} mouse forebrain exhibited an almost completely preserved MBP expression pattern (Suppl. Fig. 5c,d). Overall, these data indicate that cell-intrinsic features associated with OPC developmental heterogeneity underlie the alternative postnatal fates of forebrain *Cit-k KO* dOPCs and vOPCs.

We thank the Reviewer for her/his comment that prompted us to perform novel experiments aimed at clarifying this crucial point.

2. The authors claimed that there is more cell apoptosis in dorsal OPCs than ventral OPCs. However, they did not present *in vivo* data in the brain e.g. by immunocytochemistry.

In the former version of the manuscript, Fig.4a showed data concerning *in vivo* OPC apoptotic rates at dorsal vs. ventral sites. Representative images of Olig2⁺/cCasp3⁺ cells were in Suppl. Fig. 3c,d. For the sake of clarity, the graph in Fig.4a is now accompanied by 2 *in vivo* representative images (NG2⁺/cCasp3⁺ and Olig2⁺/cCasp3⁺ cells) in Fig. 4b,c.

3. The senescent phenotype of vOPCs in Fig 4 was not confirmed in the brain of *Cit-k* KO mice.

We now provide representative images of Sox10⁺ oligodendroglial cells displaying SA-βGAL positivity in *Cit-k* KO and Sox10^{Cre};Cit-k^{fl/fl} mouse striatum in Fig.4g-g' and Fig.5d, respectively. To strengthen the characterization of *Cit-k* KO vOPC senescent state, western blotting and qRT-PCR have been used to quantify other markers of cell senescence (i.e. p21, p16, p27 proteins and *p21*, *p16*, *Glb1*, *Ecrg4* transcripts) in OPCs immediately after their MACS-isolation from mouse brain (Fig.4j-l). In addition, we exploited BrdU incorporation *in vivo* to show the reduced mitotic activity of vOPCs in *Cit-k* KO mouse striatum (Fig.4m).

4. In Fig 5a-d, the authors showed that *Cit-k* KO dOPCs and vOPCs display the same expression levels of DNA damage response marker γH2AX, however, DNA damage response-induced ROS between dOPCs and vOPCs is different (Fig. 5c,d). The measurement of DHE is based on immunofluorescence intensity, which is not reliable for quantification. To assess ROS levels with an independent and objective method, we have now included data obtained from a quantitative Nitroblue tetrazolium (NBT) reduction assay (Choi et al., 2006 doi:10.1080/15321810500403722), where formazan particles, produced in presence of intracellular O₂⁻, were dissolved with KOH and DMSO and the absorbance was determined using a microplate reader. Consistent with DHE analysis, with this alternative quantitative method, *Cit-k* KO dOPCs were found to accumulate higher levels of ROS (i.e. showed higher absorbance values) compared to vOPCs (Fig.6e).

5. There are no experimental studies to address the mechanisms for differential ROS responses between dOPCs and vOPCs in *Cit-k* KO mice.

In response to this criticism, to validate the role of NRF2 suppression in determining *Cit-k* KO dOPC depletion we exploited a pharmacological approach and used dimethyl fumarate (DMF), a positive modulator of NRF2 stability and transcriptional activity (Scuderi et al., 2020) in an *in vitro* survival assay. We found that DMF significantly increased *Cit-k* KO dOPC survival, obtaining the same rescue detected when cells were treated with the antioxidant agent N-acetylcysteine (NAC). These data are included in Fig. 6m.

As regards lineage-associated mechanisms operating upstream to the diverse regulation of NRF2 in dOPCs vs vOPCs, we hypothesize that these include epigenetic factors, such as miRNAs or lncRNAs acting directly on NRF2 or on its interactors, that can be inherited from ventricular precursors as it was shown for neuroblasts (Olguín-Albuérne and Morán 2018, Antioxid Redox Signal) (discussion, lines 468-479). The identification of these epigenetic factors, possibly via an unbiased -omic approach, and their manipulation will require an extensive work that we reasoned to not include in the current study.

6. *Cit*-kinase KO results in DNA damage responses and oxidative stress in OPCs. It is unclear whether these events are directly regulated by *Cit* kinase or secondary to the *Cit*-kinase loss-induced cell death responses.

Recent studies (Bianchi et al., 2017 Cell Rep; Pallavicini et al., 2018 Cancer Res; Pallavicini et al., 2020 Cancers) showed that accumulation of DNA damage immediately follows *Cit-k* knockdown (i.e. 24h and 72h after knockdown induction), and that *Cit-k* knockdown reduces nuclear expression of the DNA repair protein RAD51 and impairs homologous recombination. Moreover, Bianchi and colleagues (2017 Cell Rep) previously showed the persistence of γH2AX accumulation in the double *Cit-k* KO/*p53*-KO where apoptosis was inhibited. Together, these findings indicate that accumulation of DNA damage is a direct consequence of *Cit-k* loss, rather than a secondary outcome of the activation of the apoptotic pathway. The activation of a robust antioxidant response in *Cit-k* KO vOPCs, showing low levels of apoptosis compared to dOPCs, indicates that ROS production is not directly linked to the activation of the apoptotic pathway. Rather, it may be part of the response to DNA damage (Kang et al., 2012 Cell Death Dis; Srinivas et al., 2018 Redox Biol). This is now clarified at lines 295 and 419-420.

7. qPCR measurement of *Nrf2* expression did not show the difference between dorsal OPCs and ventral OPCs in Fig. 5h, however, in figure 5k, the authors used the immunostaining intensity to quantify the NRF2 expression level. This kind of quantification is not convincing. They should do western blot analysis for the sorted cells.

According to the Reviewer's suggestion, we performed Western Blot analyses on whole-cell protein lysates of MACS-sorted *Cit-k* KO d/vOPCs. Consistent with data obtained by quantifying NRF2 immunolabeling,

despite *Nrf2* mRNA was upregulated in both *Cit-k* KO dOPCs and vOPCs, NRF2 protein levels were significantly lower in *Cit-k* KO dOPCs compared to vOPCs (Fig. 6f,g).

8. In the Figs 6 and 7, it is intriguing that NAC treatment rescued the OPC loss phenotype of *Cit-k* KO mice. The authors used organotypic culture but did not show if there is any increase of mature oligodendrocytes in the treated brain. In addition, they did not show if NAC treatment could rescue DNA damage responses in OPCs caused by *Cit-k* KO.

Following the Reviewer's suggestion, we have now included representative images of MBP (Fig. 7h) and CC1 (Suppl. Fig. 8b) immunolabelings in distinct regions (corpus callosum, striatum, anterior commissure, cerebellar white matter) of WT, *Cit-k* KO and NAC-treated *Cit-k* KO mouse brain. Both markers of mature oligodendrocytes did not increase significantly in the treated *Cit-k* KO brains. Nevertheless, NAC reduced ROS accumulation (Fig. 7b) and DNA damage (i.e. γ H2AX foci; Fig. 7c) in dOPCs.

9. The authors did not show whether treatment with NAC inhibits ROS responses e.g. DHE in *Cit-k* KO OPCs? Does NAC treatment increase OPC survival and proliferation from *Cit-k* KO?

Following the Reviewer's suggestion, we have now included data showing a significant decrease of ROS in dOPCs acutely isolated from NAC-treated *Cit-k* KO vs. control *Cit-k* KO mice, as measured by NBT assay (Fig. 7b). NAC treatment strongly reduced the fraction of apoptotic (i.e. cCASP3⁺) OPCs in both dorsal cortex and corpus callosum (Fig. 7f) and the percentage of mitotic (i.e. PH3⁺) OPCs in the corpus callosum (Fig. 7g), suggesting a promotion of differentiation. However, within the time window of treatment, OPC differentiation (i.e. expression of mature markers, such as CC1 and MBP protein, or *Gpr17*, *Plp1*, *Mbp* transcripts) was not restored in *Cit-k* KO mouse brain (Fig. 7h; Suppl. Fig. 8a,b). Thus, NAC treatment reduced ROS accumulation and sustained cell survival in dOPCs.

10. They isolated cells from P8 *Emx1*Cre-R26RYFP mouse dorsal cortex and ventral forebrain. *Emx1*Cre is mainly restricted to dorsal neural progenitors. It is not clear why there are many *Emx1*CreGFP⁺ cells in vOPCs (Fig 7d). MACS sorting with PDGFR α antibody could also isolate other cell types like NG2⁺ pericytes. The authors did not validate that the isolated NG2⁺ cells are OPCs.

Consistent with Kessaris et. al 2006, we found YFP⁺ OPCs in the medial septum and periventricular striatum of *Emx1*^{Cre};R26^{YFP} forebrain slices, thus we assumed that YFP⁺ cells within vOPCs corresponded to septal and striatal OPCs. Nevertheless, also in view of this Reviewer's criticism, we have now removed data obtained in OPCs isolated from the dorsal vs. ventral forebrain of *Emx1*^{Cre};R26R^{YFP} mice, and assessed the link between OPC fate and developmental origin by studying the relative survival rate of YFP⁺ vs. YFP-negative OPCs isolated from the entire *Emx1*^{Cre};R26R^{YFP} mouse forebrain (Fig. 8d-g). Also in this setting, where dorsally- and ventrally-derived OPCs are cultured together, *Emx1*-derived OPCs showed a higher vulnerability to cisplatin (Fig. 8g). Also, purity of the MACS-selected OPCs was verified by immunocytochemistry: i.e. more than 95% of the cells were NG2⁺ and about 98% of the cells showed positivity for PDGFR α and Sox10 (Suppl. Fig. 10a). As a further evidence of the robustness of the MACS sorting and of the virtually exclusive oligodendroglia identity of the isolated cells, when we MACS sorted cells from Sox10^{Cre};R26R^{YFP} mouse forebrain, the obtained cell population was 100% YFP⁺ and NG2⁺ (Suppl. Fig. 10b).

11. The authors indicate that p53 is correlated with NRF2 stability. The authors showed p21 upregulation in ventral OPCs, they should show p53 and phospho-p53 expression in *Cit-k* KO brain. Does NAC treatment alter p53 and p21 expression in the *Cit-k* KO brain?

A previous study showed that p53 and phospho-p53 were significantly upregulated in *Cit-k* KO brain (Bianchi et al., 2017 Cell Rep). Following the Reviewer's suggestion, we now include data showing that NAC treatment did not alter p53 and p21 mRNA levels in the KO tissue (Suppl. Fig. 8a). Yet, since new analyses showed that p53 and phospho-p53 did not vary in *Cit-k* KO dOPCs vs. vOPCs, we removed the section of the discussion proposing distinct p53 levels or activation as candidate mechanisms upstream to NRF2 suppression in dOPCs.

12. In the Suppl. Fig 2 the authors used NG2CreERTM;R26RYFP to conclude that embryonically generated OPCs were not affected by *Cit-k* KO. However, in the text (Page 7 line 158-163, also fig 3q), they stated that embryonically generated OPCs decreased in *Cit-k* KO. Authors should clarify the discrepancy. To respond to this comment and to other criticisms raised by Reviewer#3, we quantified the number of NG2⁺ OPCs distributed along the dorso-ventral axis of *Cit-k* KO vs. WT mouse brain at E14. This number was similar in the two genotypes (Suppl. Fig. 2a-g). Moreover, fate mapping of the progeny of NG2⁺ cells tagged at E14 in NG2CreERTM;R26R^{YFP} x WT/ *Cit-k* KO mice (Fig. 3o-q') confirmed a comparable OPC expansion in both *Cit-k* KO and WT POA (Fig. 3r). In more dorsal parts of the *Cit-k* KO forebrain (i.e. basolateral cortex, striatum and dorsal cortex) the number of YFP⁺ OPCs displayed a trend for a decrease compared to WT (Fig. 3r),

suggesting a local reduction of OPC expansion/survival or a defective OPC migration to these areas. A defective colonization of the dorsal forebrain by early born OPCs was also supported by the reduced number of YFP-negative OPCs in P2 $Emx1^{Cre};R26R^{YFP} \times Cit-k$ KO mice compared to WT (Fig. 3n). Thus, embryonically generated OPCs are produced and persist in the ventral telencephalon of $Cit-k$ KO, while their colonization of the dorsal forebrain is hampered. These findings and interpretations are now reported at lines 155-172.

Reviewer #3 (Remarks to the Author):

This paper by Boda and colleagues entitled “Citron-kinase deletion unveils inherent molecular and functional heterogeneity in dorsal and ventral oligodendrocyte precursor cells of the mouse forebrain” investigates the effect of genetic deletion of Citron-kinase (Cit-k), a mouse model of human microlissencephaly associated with perinatal lethality in humans and mice. Cit-k has previously been demonstrated to be required for mitosis, organization of the midbody in late cytokinesis and in DNA damage control (doi:10.1242/jcs.200253). Boda and colleagues reveal that Cit-k KO mice exhibit a dramatic loss of pre-myelinating OLs, oligodendrocytes and myelin during postnatal development. They provide evidence of a generalized reduction in the density of OPCs in the Cit-k KO forebrain. The reduction in OPC density was greater for dorsal OPCs than ventral OPCs, with the former exhibiting a progressive reduction from P3 to P14, unlike ventral OPCs which maintained the same density at both time-points. The regional differences in OPC response to Cit-k deletion are ascribed to cell autonomous differences in the response of dorsal and ventral OPCs to DNA-damage induced oxidative stress that are dependent on the developmental origin of the OPCs. They provide evidence that dorsal OPCs that derive from Emx1-progenitors exhibit greater vulnerability to Cit-k deletion and undergo apoptotic cell death. By contrast, ventrally-derived OPCs were more resilient to DNA damage, undergoing senescence rather than apoptosis, which was associated with the upregulation of Nuclear factor erythroid 2-related factor 2 (NRF2), a key mediator of antioxidant defense. They also demonstrate that antioxidant supplementation to Cit-K KO mice confer resistance of dorsal OPCs to DNA damage.

The interest in this work lies in the molecular and functional differences in OPCs of differing developmental origin. The study builds on prior work demonstrating heterogeneity in the function of OPCs that derive from different progenitor cell populations, here demonstrating that dorsal and ventral OPCs differ in their response to DNA damage. The intrinsic differences in dorsal and ventral OPCs to DNA damage has implications for understanding the susceptibility of dorsal OPCs to perinatal hypoxia-ischemia and ageing.

Overall, the study is generally well conducted and the results are important, but this reviewer has a number of concerns that the authors should address to provide more compelling evidence for their conclusions.

Specific concerns:

1) The authors conclude that “dorsal oligodendrogenesis is not affected by loss of Cit-k” (lines 149-150) and that “normal numbers of OPCs are generated postnatally in the dorsal cortex but undergo a progressive depletion” (lines 164-165). This is based on density measures of fate-mapped OPCs. To reach this conclusion the authors should clarify the labeling efficiency of stem/progenitor cells in the lateral ventricles of WT and Cit-k KO mice using their genetic labeling approaches. Moreover, presenting the results as a density measure alone is misleading since the Cit-k KO brain is significantly smaller than WT. The authors should also provide the total number of neural stem/progenitor cells genetically tagged and the total number cell of YFP+/PDGFR α + OPCs across the corpus callosum and cerebral cortex. The combination of a genetic approach with EdU pulse/chase labeling would facilitate this analysis.

Following the Reviewer’s suggestion, in the fate mapping analyses exploiting RV-GFP and Tat-Cre, in addition to present OPC quantifications as absolute numbers, we now report the absolute number of the precursors targeted in the dSVZ along with the ratio between the number of targeted parenchymal OPCs (derivatives) and the number of dSVZ targeted cells. With both lineage tracing approaches, although the absolute number of the targeted dSVZ precursors was decreased in *Cit-k* KO compared to WT, no significant change in the number of their parenchymal NG2⁺ dOPC derivatives was observed (Fig. 3d,i). Consistently, an increased ratio of parenchymal GFP/YFP⁺ dOPCs to their dSVZ ancestors was observed in KO mice (Fig. 3e,j). We now also show absolute numbers of YFP positive/negative cortical OPCs in Emx1^{Cre};R26R^{YFP} x WT/*Cit-K* KO (Fig. 3n). Also in this model, the number of dorsally-derived OPCs did not differ in the two genotypes. Overall, these data show that dorsal oligodendrogenesis is not negatively affected in *Cit-K* KO (lines 130-154).

2) Line 145-146: Please clarify the meaning of the statement “90% of YFP+/PDGFR α + OPCs residing in the dorsal cortex/CC of both WT and Cit-k KO mice”. The figure illustrates extensive labelling of cells along the walls of the lateral ventricle but little labeling in the CC or cortex.

We agree with the Reviewer that the sentence was unclear. For the sake of clarity, we have now modified the sentence in “with about 90% of parenchymal YFP+/PDGFR α + OPCs residing in the dorsal cortex/CC of both WT and *Cit-k* KO mice”. With this sentence we wanted to emphasize that most OPCs generated postnatally from SVZ precursors eventually populate the dorsal forebrain, rather than other regions.

3) The analysis of ventral OPC density is insufficient to reach the conclusion that vOPC genesis and distribution is largely unaffected in the Cit-k KO. To support their conclusion, the authors refer to Supplementary Figure 2 which shows that

there is no significant difference in the percentage of YFP+NG2+ OPCs in dorsal, lateral and ventral forebrain in NG2CreERTM;R26RYFP WT and Cip-k KO mice. The genetic labelling approach is not adequately analysed to form a clear conclusion. Most importantly, they do not support the conclusion that there are differential effects on OPCs in dorsal versus ventral forebrain. The analysis also does not take into account differences in OPC density at the time of tamoxifen treatment and the number or density of OPCs at the time of analysis (P10).

Following the Reviewer's suggestion, to corroborate the conclusion that vOPC generation is largely preserved in *Cit-k KO* mouse forebrain, we have quantified OPCs in E14 *Cit-k KO* vs. WT forebrain and showed that the number of NG2+ cells distributed along the dorso-ventral axis of *Cit-k KO* mouse brain was similar to that of WT samples (Suppl. Fig. 2a-g). Thus, at E14, tamoxifen could drive Cre expression in equivalent numbers of cells in NG2CreERTM;R26RYFP x WT/ *Cit-k KO* mice. Moreover, we quantified the absolute number of the progeny of NG2+ cells tagged at E14 in NG2CreERTM;R26RYFP x WT/ *Cit-k KO* mice at distinct dorso-ventral sites (Fig. 3o-q'). This analysis confirmed a comparable OPC expansion in both *Cit-k KO* and WT POA at P10 (Fig. 3r). In more dorsal parts of the *Cit-k KO* forebrain (i.e. basolateral cortex, striatum and dorsal cortex) the number of YFP+ OPCs displayed a trend for a decrease compared to WT (Fig. 3r), suggesting a local reduction of OPC expansion/survival or a defective OPC migration to these areas. A defective colonization of the dorsal forebrain by early born OPCs was also supported by the reduced number of YFP-negative OPCs in P2 *Emx1Cre*;R26RYFP x *Cit-k KO* mice compared to WT (Fig. 3n). Together, these findings support the view that at least the first wave of vOPCs is correctly generated in *Cit-k KO*. Yet, *Cit-k* deletion impacts the ability of embryonically generated OPCs to populate the dorsal forebrain (lines 155-168).

Fate mapping analysis in *Nkx2.1Cre*;R26RYFP;CIT-k^{fl/fl} (first wave from ventral MGE/AEP/POA precursors) also showed that *Cit-k* loss did not impact the number of ventrally-derived OPCs in the POA at P14-P16 (Fig. 5g,j; lines 258-275).

4) To authors state that "dOPCs underwent apoptosis with higher frequencies compared to vOPCs, despite uniform cytokinesis defects across both populations". Although the authors provide photomicrographs illustrating incomplete binucleated OPCs in both dorsal and ventral forebrain, the abundance relative to mononucleated OPCs is not quantified. Multinucleated OPCs of the type characterized would also be expected of normal cytokinesis in WT mice. Quantification of the percentage of OPCs that are binucleated in WT versus Cip-k KO in both dorsal and ventral forebrain would be required to support any conclusion about cytokinesis defects.

Following the Reviewer's suggestion, we quantified the abundance of multinucleated OPCs at distinct sites along the dorso-ventral axis of WT vs. *Cit-k KO* forebrain at P10. Although many WT OPCs showed two nuclei during the last phases of mitosis (i.e. during telophase, cytokinesis) and immediately after cell division (when DNA appeared grainy and condensed as shown in Boda et al., 2015 *Glia*, doi: 10.1002/glia.22750, Fig. 6F-K), multinucleated interphasic OPCs were virtually absent in the WT forebrain. In *Cit-k KO* forebrain, instead, multinucleated OPCs have been found. Their fraction differed at distinct sites, with POA OPCs showing the lowest percentage of multinucleation (Suppl. Fig. 3c,d). However, these differences showed no correlation with the apoptotic fractions observed in distinct areas (Simple Linear Regression, $R^2=0.1981$, $P=0.1557$, the slope was not significantly different to 0; Suppl. Fig. 3e), e.g. CC and striatal OPCs showed similar percentages of multinucleated cells, but significantly different apoptotic fractions. Complementarily, striatal OPCs showed a 2-fold increase in the percentage multinucleated cells compared to POA OPCs, but a similarly low rate of apoptosis.

These findings, together with the observations that cCASP3 colocalized with both single and multi-nucleated dOPCs (Fig. 4b,c) and that OPCs were almost completely depleted in P14 *Cit-K KO* dorsal cortex, where multinucleated cells accounted only for a third of the entire OPC population at P10, support the view that cell death in *Cit-k KO* OPCs is not a direct consequence of incomplete cytokinesis, and that multinucleation is an overlapping phenotype emerging as a consequence of *Cit-k* loss. This concept is now included at lines 189-201.

5) Lines 220-222: The authors state that "the acquisition of senescent properties in *Cit-k KO* vOPCs depended on cell-intrinsic factors, rather than on environmental senescence-inducers" is not supported by any analysis of senescent markers in the *Cit-k KO* recipient of GFP+ wildtype telencephalic cells. This would require an examination of a senescence marker such as SA- β GAL in GFP+ cells in the ventral forebrain.

We agree with the Reviewer. Yet, for technical reasons, we were not able to combine SA- β GAL histological labeling with anti-GFP immunostaining in slices. Thus, we indirectly assessed the proliferative/migratory ability of GFP+ OPCs in the ventral forebrain by quantifying their post-transplantation expansion in the striatum/thalamus (Suppl. Fig. 4c). Here, grafted OPCs significantly increased in number, colonized a large area of the tissue, and even proceeded along the lineage, displaying premyelinating markers and alignment of processes to axons (Suppl. Fig. 4d and insets). Thus, WT OPCs did not exhibit major defects when grafted

in the ventral *Cit-K KO* forebrain, although they appeared to amplify less compared to GFP⁺ cells in the dorsal cortex (Suppl. Fig. 4c). This could be due to their competition with the abundant population of resident vOPCs. Yet, we agree that these observations do not completely exclude the presence of environmental senescence inducers. Therefore, in this revised study, we addressed the contribution of cell intrinsic vs. environmental factors by examining an oligodendroglia-specific *Cit-k* mutant (i.e. Sox10^{Cre};Cit-k^{fl/fl} mice) and showed expression of senescence markers - including SA-βGAL - in Sox10^{Cre};Cit-k^{fl/fl} vOPCs (Fig. 5d,e). These data indicate that the acquisition of senescent properties in vOPCs upon *Cit-k* loss depended on cell-intrinsic factors.

6) Figure 5. The authors should provide the micrographs of WT P10 OPCs stained with gamma-H2AX. Following the Reviewer's suggestion, we have now included micrographs of WT P10 OPCs stained with γH2AX (Fig. 6a of the revised manuscript).

7) The assessment of apoptosis by activated caspase-3 labeling was performed only on OPCs isolated from dorsal cortex and corpus callosum of Cit-KO mice. To argue that dOPCs are more vulnerable to apoptosis a comparative analysis of apoptosis in dOPCs and vOPCs should be conducted. In the former and current version of the manuscript, Fig.4a shows data concerning in vivo OPC apoptotic rates at dorsal (dorsal cortex and corpus callosum) vs. ventral (striatum and POA) sites. Since also Reviewer#2 raised a similar criticism about that part of data, for the sake of clarity we now modified the type of plot illustrating the data and opted for a "dorsal to ventral" histogram representation. We also moved the representative images from the supplementary material to the principal figure (Fig. 4b,c).

8) The referring figure number needs to be added to Supplementary Table 1 row 1. ANCOVA analysis refers to data included in the text (lines 128-130; not shown in a figure).

Reviewers' comments:

Reviewer #1 (Remarks to the Author):

The authors have adequately addressed all of my concerns

Reviewer #2 (Remarks to the Author):

The authors have addressed my major concerns. I only have a minor question:

Since the forebrain size of *Emx1Cre;R26RYFP;Cit-kfl/fl* mice is reduced, authors state that the MBP expression pattern is preserved, could the authors quantify the number of oligodendrocytes e.g., CC1+, and examine changes in other cell types such as astrocytes and neurons in the mutant mice?

Reviewer #3 (Remarks to the Author):

The revised paper by Boda and colleagues has been much improved since its original submission. The current version of the manuscript incorporates a significant body of additional experiments that further support the conclusions of the authors and address the main concerns that this reviewer raised. In particular, the study now provides compelling evidence that dorsal and ventral OPCs exhibit divergent responses to DNA damage and oxidative stress, which results in them adopting either an apoptotic or senescent fate, respectively. The authors identify key molecular players (Nrf2, Puma) within the oxidative stress pathway that help explain these divergent responses. They also demonstrate that mediators of cellular senescence (including p21 and p16) are expressed at different levels in dorsal versus ventral OPCs in the context of *Cit-k* deletion. Together, the work highlights that lineage-specific differences in OPC responses to oxidative stress can be revealed through either genetic or environmental induction. The authors also provide new experimental data using conditional deletion of *Cit-k* in dorsal or ventral OPCs and transplantation of WT cells into a *Cit-k* KO brain, demonstrating the cell-intrinsic nature of these divergent responses. As such, the study builds on existing literature showing divergent responses of ventrally and dorsally-specified OPCs, which reflect cell-intrinsic differences between these two distinct OPC lineages. This well-executed and thorough study expands our understanding of the nature of the heterogeneity of OPCs in the brain and the implications of such diversity.

Response to Reviewers

Reviewer #1 and Reviewer #3 did not express concerns. We thank them for their supportive statements.

Reviewer #2 (Remarks to the Author):

The authors have addressed my major concerns. I only have a minor question: Since the forebrain size of *Emx1^{Cre};R26^{YFP};Cit-k^{fl/fl}* mice is reduced, authors state that the MBP expression pattern is preserved, could the authors quantify the number of oligodendrocytes e.g., CC1+, and examine changes in other cell types such as astrocytes and neurons in the mutant mice?

We thank the Reviewer for this comment that prompted us to perform new immunostainings and quantifications to better characterize the phenotype of juvenile *Emx1^{Cre};R26^{YFP};Cit-k^{fl/fl}* mutants. Specifically, we found a 50% reduction of cortical BLBP⁺ astrocytes in both superficial (I-III) and deep (IV-VI) layers of the cortex (Supplementary Figure 5h,i) and a 30% reduction of cortical CC1⁺ mature oligodendrocytes in the deep (IV-VI) layers of the cortex (Supplementary Figure 5j,k). Finally, Nissl staining was used to highlight the reduction of cortical thickness and provide representative images of neuronal aspect and layering (i.e. to show that despite a certain disorganization of the cortical layers, pyramidal neurons could be detected in *Emx1^{Cre};R26^{YFP};Cit-k^{fl/fl}* mouse cortex; Supplementary Figure 5g).

These findings are now included in the manuscript (lines 265-270), where we toned down our former statement about MBP expression: “In line with this interpretation, *Emx1^{Cre};R26^{YFP};Cit-k^{fl/fl}* mouse dorsal cortex exhibited a relatively preserved MBP expression pattern (at least in the deeper layers; Suppl. Fig. 5c,d), although showing a 30% reduction of CC1⁺ mature oligodendrocytes (Suppl. Fig. 5j,k). However, cortical size was decreased (Suppl. Fig. 5c,d,g) and BLBP⁺ astrocyte reduced in number (Suppl. Fig. 5h,i), consistent with a global defect in dorsal VZ/SVZ derivatives in *Emx1^{Cre};R26^{YFP};Cit-k^{fl/fl}* mice.”.